# Pathogenic mutations in the chromokinesin KIF22 disrupt anaphase chromosome segregation

**Alex F Thompson[1], Patrick R Blackburn[2,3], Noah S Arons[1], Sarah N Stevens[1], Dusica Babovic-Vuksanovic[2,4], Jane B Lian[5], Eric W Klee[6], Jason Stumpff[1]***

[1]Department of Molecular Physiology and Biophysics, University of Vermont, Burlington, United States; [2]Laboratory Medicine and Pathology, Mayo Clinic, Rochester, United States; [3]Pathology, St. Jude Children's Research Hospital, Memphis, United States; [4]Clinical Genomics, Mayo Clinic, Rochester, United States; [5]Department of Biochemistry, University of Vermont, Burlington, United States; [6]Biomedical Informatics, Mayo Clinic, Rochester, United States

**Abstract** The chromokinesin KIF22 generates forces that contribute to mitotic chromosome congression and alignment. Mutations in the α2 helix of the motor domain of KIF22 have been identified in patients with abnormal skeletal development, and we report the identification of a patient with a novel mutation in the KIF22 tail. We demonstrate that pathogenic mutations do not result in a loss of KIF22's functions in early mitosis. Instead, mutations disrupt chromosome segregation in anaphase, resulting in reduced proliferation, abnormal daughter cell nuclear morphology, and, in a subset of cells, cytokinesis failure. This phenotype could be explained by a failure of KIF22 to inactivate in anaphase. Consistent with this model, constitutive activation of the motor via a known site of phosphoregulation in the tail phenocopied the effects of pathogenic mutations. These results suggest that the motor domain α2 helix may be an important site for regulation of KIF22 activity at the metaphase to anaphase transition. In support of this conclusion, mimicking phosphorylation of α2 helix residue T158 also prevents inactivation of KIF22 in anaphase. These findings demonstrate the importance of both the head and tail of the motor in regulating the activity of KIF22 and offer insight into the cellular consequences of preventing KIF22 inactivation and disrupting force balance in anaphase.

*For correspondence: jstumpff@uvm.edu

Competing interest: The authors declare that no competing interests exist.

## Editor's evaluation

This article analyzes the mechanism of human pathogenicity linked to point mutations in the chromokinesin Kid/Kif22, that cause abnormal skeletal development. The authors show the mutations do not cause a loss of function. Instead, they are dominant negative and fail to inactivate the Kif22 motor, resulting in appropriate force generation of the motor during anaphase. This work highlights that the loss of regulation of kinesin motors in mitosis can disrupt cell division at the cellular scale and human pathogenesis.

## Introduction

Mitosis requires mechanisms that mechanically control chromosome movements to ensure equal segregation of chromosomes to daughter cells. Forces that move mitotic chromosomes are generated by microtubule dynamics within the mitotic spindle and by molecular motor proteins. The chromokinesin KIF22 (or Kid, kinesin-like DNA-binding protein) is a plus-end directed member of the kinesin-10

family (*Yajima et al., 2003*). KIF22 and its orthologs, including Nod (*Drosophila melanogaster*) (*Zhang et al., 1990*) and Xkid (*Xenopus laevis*) (*Antonio et al., 2000*; *Funabiki and Murray, 2000*; *Takagi et al., 2013*), generate forces that move chromosomes away from the spindle poles. Structurally, KIF22 contains a conserved kinesin motor domain responsible for ATP hydrolysis and microtubule binding (*Tokai et al., 1996*; *Yajima et al., 2003*), a second microtubule-binding domain in the tail (*Shiroguchi et al., 2003*), a predicted coiled-coil domain (*Shiroguchi et al., 2003*), and a C-terminal DNA binding domain, which includes a helix-hairpin-helix motif (*Tokai et al., 1996*; *Figure 1A*). Precisely how KIF22's force generating activity is regulated in mitotic cells and how this regulation contributes to spindle function and cell viability remains incompletely understood.

In interphase, KIF22 localizes to the nucleus (*Levesque and Compton, 2001*; *Tokai et al., 1996*). As cells enter mitosis, chromosomes condense and KIF22 binds along chromosome arms (*Levesque and Compton, 2001*; *Tokai et al., 1996*). In prometaphase, chromosomes must congress and align at the center of the spindle. The interactions of the KIF22 motor domain with spindle microtubules and the KIF22 tail with chromosome arms allow the motor to generate polar ejection forces (*Bieling et al., 2010*; *Brouhard and Hunt, 2005*), which push the arms of chromosomes away from the spindle poles and toward the center of the spindle (*Marshall et al., 2001*; *Rieder and Salmon, 1994*; *Rieder et al., 1986*), contributing to chromosome congression in prometaphase (*Iemura and Tanaka, 2015*; *Levesque and Compton, 2001*; *Wandke et al., 2012*), as well as chromosome arm orientation (*Levesque and Compton, 2001*; *Wandke et al., 2012*). In metaphase, polar ejection forces also contribute to chromosome oscillation and alignment (*Antonio et al., 2000*; *Funabiki and Murray, 2000*; *Levesque and Compton, 2001*; *Levesque et al., 2003*; *Stumpff et al., 2012*; *Takagi et al., 2013*; *Tokai-Nishizumi et al., 2005*). Purified KIF22 is monomeric (*Shiroguchi et al., 2003*; *Yajima et al., 2003*), and the forces generated by KIF22 on chromosomes arms may represent the collective action of many monomers. In anaphase, KIF22 is inactivated to reduce polar ejection forces and allow chromosomes to segregate toward the spindle poles (*Soeda et al., 2016*; *Su et al., 2016*; *Wolf et al., 2006*).

The generation of polar ejection forces by KIF22 is regulated by the activity of cyclin-dependent kinase 1 (CDK1)/cyclin B, which is high in prometa- and metaphase, and drops sharply at the metaphase to anaphase transition when cyclin B is degraded (*Hershko, 1999*; *Morgan, 1995*). KIF22 is phosphorylated by CDK1/cyclin B at T463, a residue in the tail of the motor between the second microtubule-binding and coiled-coil domains. Phosphorylation of T463 is required for polar ejection force generation in prometa- and metaphase, and dephosphorylation of T463 is necessary for the suspension of polar ejection forces to allow chromosome segregation in anaphase (*Soeda et al., 2016*). Although a reduction of polar ejection forces in anaphase is a necessary step for proper anaphase chromosome segregation, it is not clear how this contributes to a shift in force balance within the spindle at the metaphase to anaphase transition. Furthermore, while several regions of the KIF22 tail are known to contribute to KIF22's inactivation as cells transition to anaphase, how motor activity is downregulated has not been resolved. Phosphoproteomic studies have identified sites of phosphorylation within KIF22's α2 helix (*Kettenbach et al., 2011*; *Olsen et al., 2010*; *Rigbolt et al., 2011*), suggesting this region, in addition to the tail, may also be important for the regulation of motor activity.

The study of pathogenic mutations can often provide insight into the regulation and function of cellular proteins. Mutations in KIF22 cause the developmental disorder spondyloepimetaphyseal dysplasia with joint laxity, leptodactylic type (SEMDJL2, also referred to as Hall Type or lepto-SEMDJL) (*Boyden et al., 2011*; *Min et al., 2011*; *Tüysüz et al., 2015*). Four point mutations in two amino acids have been reported in SEMDJL2 patients (*Boyden et al., 2011*; *Min et al., 2011*; *Tüysüz et al., 2015*; *Figure 1A*). These mutations occur in adjacent residues P148 and R149 in the α2 helix of the KIF22 motor domain (*Figure 1B*). P148 and R149 are conserved in kinesin-10 family members across species (*Figure 1C*) and in many human members of the kinesin superfamily (*Figure 1D*). However, no pathogenic mutations in the homologous proline or arginine residues have been recorded in OMIM (Online Mendelian Inheritance in Man; https://omim.org/). All identified patients are heterozygous for a single mutation in KIF22. Mutations in KIF22 dominantly cause SEMDJL2, and patients with both de novo and inherited mutations have been identified (*Boyden et al., 2011*; *Min et al., 2011*).

Although KIF22 mRNA is expressed throughout the body (Human Protein Atlas; http://www.proteinatlas.org; *Uhlén et al., 2015*), the effects of these mutations are largely tissue-specific, and

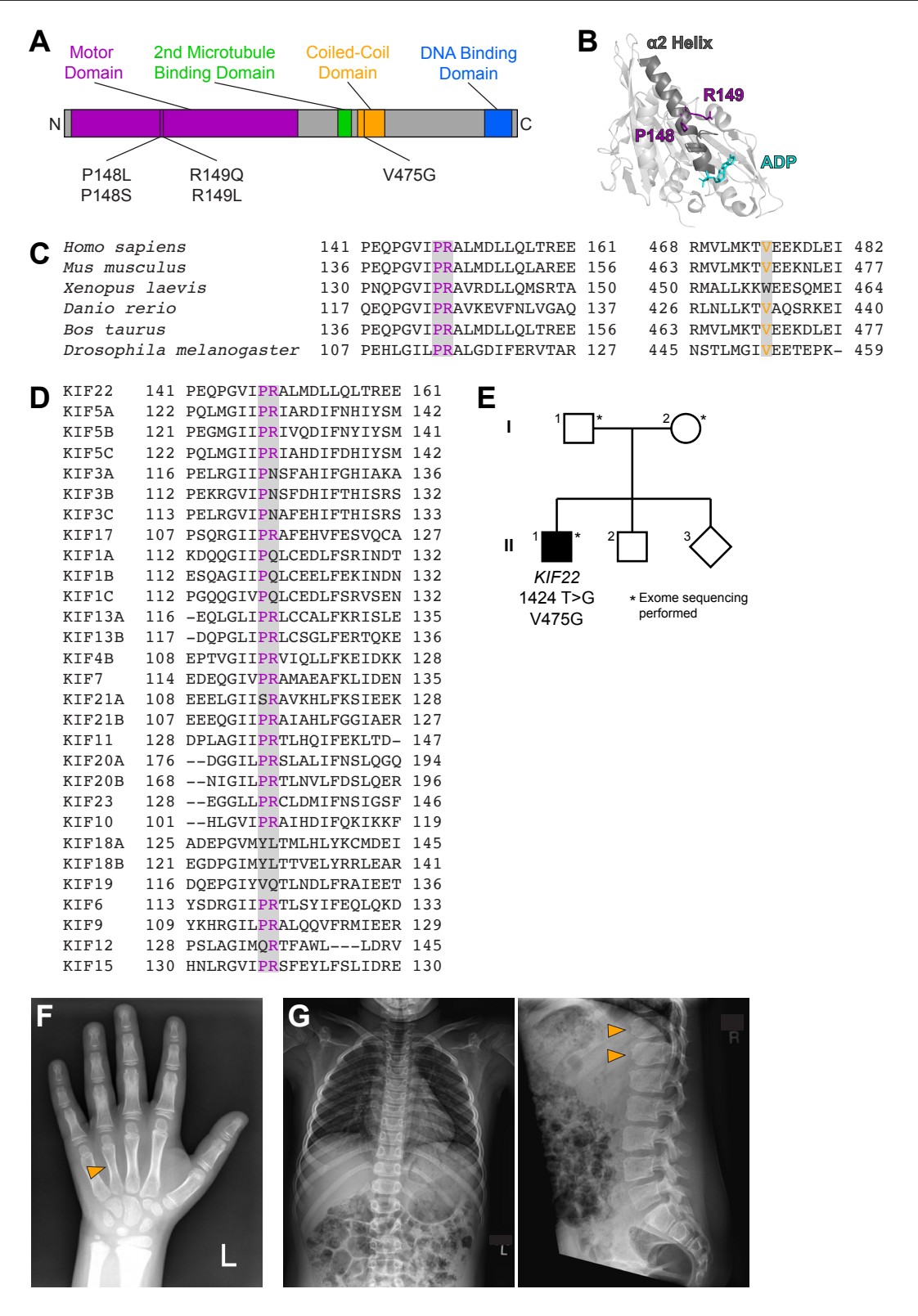

**Figure 1.** Identification of a novel pathogenic mutation in the tail of KIF22. (**A**) Schematic of the domains of KIF22 with pathogenic mutations in the motor domain (magenta) and coiled-coil domain (yellow) indicated. (**B**) Location of amino acids P148 and R149 in the α2 helix of the KIF22 motor domain (PDB 6NJE). (**C**) Alignment of amino acid sequences of kinesin-10 family members to assess conservation of motor domain (P148 and R149, left) and coiled-coil domain (V475G, right) residues across species. (**D**) Alignment of amino acid sequences of human kinesin motors to assess conservation

*Figure 1 continued*
of motor domain residues across the kinesin superfamily. For (**C, D**), alignments were performed using Clustal Omega. (**E**) Pedigree identifying the de novo V475G (1424T>G) mutation. (**F**) Radiograph of the patient's hand, posteroanterior view. Arrowhead indicates mild foreshortening of the fourth metacarpal. (**G**) Radiographs of the patient's spine. Left: anteroposterior view, right: lateral view. Arrowheads indicate 'bullet-shaped' lower thoracic vertebrae.

the development of the skeletal system is most affected in SEMDJL2 patients. A primary symptom of SEMDJL2 is short stature, resulting from shortening of both the trunk and the limbs. Additionally, patients presented with joint laxity, midface hypoplasia, scoliosis, and leptodactyly, a narrowing of the fingers (*Boyden et al., 2011*; *Min et al., 2011*). In very young children with SEMDJL2, the softness of the cartilage in the larynx and trachea caused respiratory issues (*Boyden et al., 2011*). Growth plate radiology demonstrated delayed maturation of the metaphyses and epiphyses in SEMDJL2 patients, and symptoms became more pronounced as patients aged (*Tüysüz et al., 2015*). Leptodactyly, specifically, was only observed in older (young adult) patients (*Boyden et al., 2011*).

Pathogenic mutations in the KIF22 motor domain were predicted to be loss of function mutations (*Min et al., 2011*). However, KIF22 knockout in mice did not affect skeletal development. Loss of KIF22 was lethal early in embryogenesis for approximately 50% of embryos, but mice that survived past this point developed to adulthood and demonstrated no gross abnormalities or pathologies (*Ohsugi et al., 2003*). As such, the cellular mechanism by which mutations in KIF22 affect development is unknown.

Here, we characterize an additional patient with a mutation in KIF22 and assess the effect of previously reported and novel pathogenic mutations on the function of KIF22 in mitosis. We demonstrate that mutations are not loss of function mutations, and do not alter the localization of the motor or the generation of polar ejection forces in prometaphase. Instead, mutations disrupt anaphase chromosome segregation, consistent with continued KIF22 activation and consequent polar ejection force generation in anaphase. Defects in anaphase chromosome segregation affect daughter cell nuclear morphology and, in a subset of cells, prevent cytokinesis. These findings demonstrate that anaphase inactivation of KIF22 is critical for daughter cell fitness. As such, mitotic defects may contribute to pathogenesis in patients with KIF22 mutations. Additionally, we demonstrate that aberrant polar ejection force generation in anaphase primarily affects the segregation of chromosomes by limiting chromosome arm movements in anaphase A and spindle pole separation in anaphase B, offering insight into the balance of forces required for accurate chromosome segregation in anaphase. Finally, we demonstrate that mimicking phosphorylation of T158 in the α2 helix disrupts anaphase chromosome segregation, confirming that the region of the motor domain affected by SEMDJL2 mutations also contributes to the mechanism by which KIF22 is inactivated in anaphase.

## Results

### A novel mutation in KIF22 affects development

We report the identification and characterization of a patient with a novel mutation in KIF22 (*Figure 1E*). The patient is a 15-year-old male with a history of short stature, cryptorchidism and shawl scrotum, minimal scoliosis, secondary enuresis, and skin hyperpigmentation. He presented for evaluation at 9 years of age. At that time, his height was just below 3% of age, weight was at 40% of age, and BMI was 82% of age. He was noted to have relative macrocephaly, with a head circumference at 93% of age. He had a broad forehead and hypertelorism, round face, flaring of eyebrows, and ankyloglossia. He also had mild brachydactyly (*Figure 1F*). He had a history of short stature since infancy, but followed a trajectory close to the third percentile. Growth hormone and thyroid function were normal. Bone age showed a normal, age-appropriate bone maturation with normal epiphyseal ossification centers. However, skeletal survey at age of 11 years disclosed mild foreshortening of both fourth metacarpals (*Figure 1F*), mild scoliosis of 14°, as well as mild increase of the central anteroposterior diameter of several lower thoracic vertebrae with mild 'bullet-shaped' appearance, and mild posterior scalloping of the lumbar vertebrae (*Figure 1G*).

Genetic testing was performed to determine the cause of these developmental differences. Clinical whole-exome sequencing revealed two variants of uncertain significance: a maternally inherited heterozygous *SLC26A2* variant [NM_000112.3(SLC26A2): c.1046T>A (p.F349Y)] (SCV000782516.1),

**Table 1.** Predictions of the significance of the c.1424C>G, p.(V465G) KIF22 variant.

| Algorithm | Prediction |
|---|---|
| Sorting Intolerant from Tolerant (SIFT) *Vaser et al., 2016* | Deleterious: score 0.01 with scores ranging from 0 to 1 and scores below 0.05 considered deleterious |
| Polymorphism Phenotyping (PolyPhen-2) *Adzhubei et al., 2010* | Benign: score 0.437 |
| MutationTaster *Schwarz et al., 2010* | Deleterious |
| Combined Annotation Dependent Depletion (CADD) *Rentzsch et al., 2019* | Deleterious: scaled C-score 15.3800, with a score of greater than or equal to 10 indicating a deleterious substitution |
| Deleterious Annotation of Genetic Variants Using Neural Networks (DANN) *Quang et al., 2015* | Deleterious: score 0.99 with scores ranging from 0 to 1 and higher values indicating a variant is more likely to be deleterious |
| Rare Exome Variant Ensemble Learner (REVEL) *Ioannidis et al., 2016* | Benign: score 0.28 with scores ranging from 0 to 1 and scores >0.803 classified as pathogenic |

as well as a de novo heterozygous *KIF22* variant [NM_007317.3(KIF22):c.1424T>G (p.V475G)] (SCV000782515.1) (*Figure 1E*). The *SLC26A2* gene encodes the diastrophic dysplasia sulfate transporter (*Haila et al., 2001*; *Rossi and Superti-Furga, 2001*). However, results of carbohydrate-deficient transferrin testing were not consistent with a congenital disorder of glycosylation (transferrin tri-sialo/di-oligo ratio 0.07).

The c.1424T>G, p.(V475G) *KIF22* variant has not been observed previously in the Genome Aggregation Database (gnomAD). This missense variant has mixed in silico predictions of significance (*Table 1*). According to the American College of Medical Genetics 2015 criteria, the variant was classified as a variant of uncertain significance. V475 is located in the coiled-coil domain in the tail of KIF22 (*Figure 1A*). This residue is conserved in most kinesin-10 family members across species (*Figure 1C*). However, the tail domains of kinesin motors diverge in both structure and function, and as such meaningful alignments to assess the conservation of V475 across the human kinesin superfamily were not possible.

## Pathogenic mutations in KIF22 do not disrupt the localization of the motor

To assess the effect of published pathogenic mutations in the motor domain and the novel pathogenic mutation in the tail on the function of KIF22 in mitosis, we generated human cervical adenocarcinoma (HeLa-Kyoto) cell lines with inducible expression of KIF22-GFP. Treatment of these cells with doxycycline induced KIF22-GFP expression at a level approximately two- to threefold higher than the level of expression of endogenous KIF22 as measured by immunofluorescence (*Figure 2—figure supplement 1A-C*). To facilitate both overexpression of and rescue with KIF22-GFP constructs, siRNA-resistant silent mutations were introduced into exogenous KIF22 (*Figure 2—figure supplement 1D-E*). siRNA knockdown reduced levels of endogenous KIF22 by 87% (mean knockdown efficiency across HeLa-Kyoto cell lines) (*Figure 2—figure supplement 1D*). Initial experiments were performed using HeLa-Kyoto cell lines expressing each known pathogenic mutation in KIF22 (P148L, P148S, R149L, R149Q, and V475G), and a subset of experiments then focused on cells expressing one representative motor domain mutation (R149Q) or the coiled-coil domain mutation in the tail (V475G). Additionally, we generated inducible retinal pigmented epithelial (RPE-1) cell lines expressing wild-type and mutant KIF22-GFP to assess any differences between the consequences of expressing mutant KIF22 in aneuploid cancer-derived cells (HeLa-Kyoto) and genomically stable somatic cells. RPE-1 cells are human telomerase reverse transcriptase (hTERT)-immortalized (*Bodnar et al., 1998*), and metaphase chromosome spreads demonstrated that these cell lines are near-diploid, with a modal chromosome number of 46, even after selection to generate stable cell lines (*Figure 2—figure supplement 1F-G*). The expression level of siRNA-resistant KIF22-GFP in RPE-1 cell lines was approximately four- to sevenfold higher than the level of expression of endogenous KIF22 (*Figure 2—figure supplement 1H-K*), and siRNA knockdown reduced levels of endogenous KIF22 by 67% (mean knockdown efficiency across RPE-1 cell lines measured using immunofluorescence). As measurements of KIF22 depletion by immunofluorescence may include non-specific signal, this estimate of knockdown efficiency may underestimate the depletion of KIF22.

KIF22 localizes to the nucleus in interphase, and primarily localizes to chromosomes and spindle microtubules during mitosis (*Tokai et al., 1996*). KIF22-GFP with pathogenic mutations demonstrated the same localization pattern throughout the cell cycle as wild-type motor (*Figure 2A*). In all cell lines, KIF22-GFP was localized to the nucleus in interphase cells and was bound to condensing chromosomes in prophase. In prometaphase, metaphase, and anaphase mutant and wild-type KIF22-GFP localized primarily to chromosome arms, with a smaller amount of motor signal visible on the spindle microtubules. The same localization patterns were seen for mutant and wild-type KIF22-GFP expressed in RPE-1 cells (*Figure 2—figure supplement 2* A).

Since mutations did not grossly disrupt localization of KIF22-GFP, fluorescence recovery after photobleaching (FRAP) was used to compare the dynamics of mutant and wild-type KIF22 localization. In interphase nuclei, KIF22-GFP signal recovered completely 220 s after bleaching (97±3% of intensity before bleaching, mean ± SEM), indicating a dynamic pool of KIF22-GFP (*Figure 2B* and *Figure 2—figure supplement 2B*). Similar high recovery percentages were also measured in interphase nuclei of cells expressing KIF22-GFP R149Q and KIF22-GFP V475G (100±6% and 103±7% at 220 s, respectively) (*Figure 2E and H*). In contrast, KIF22-GFP recovery was minimal in cells bleached during metaphase and anaphase. Immediately after bleaching KIF22-GFP in metaphase cells, intensity was reduced to 18±3% of initial intensity, and intensity had recovered to only 25±3% after 220 s (*Figure 2C* and *Figure 2—figure supplement 2B*). In anaphase, KIF22-GFP intensity immediately after bleaching was 17±2% of initial intensity, and intensity recovered to 35±6% of initial intensity after 220 s (*Figure 2D* and *Figure 2—figure supplement 2B*). This limited recovery indicates that KIF22 stably associates with mitotic chromosomes. Pathogenic mutations did not change these localization dynamics; recovery percentages in mitosis were also low in cells expressing KIF22-GFP R149Q (32±3% of initial intensity in metaphase 220 s after bleaching, 39±6% in anaphase) (*Figure 2F and G*) and KIF22-GFP V475G (29±2% of initial intensity in metaphase, 35±6% in anaphase) (*Figure 2I and J*; *Video 1*). These data indicate that pathogenic mutations do not alter the localization of KIF22 to chromosomes and spindle microtubules, and do not alter KIF22 localization dynamics in interphase, metaphase, or anaphase.

## Mutations do not reduce polar ejection forces

In prometaphase and metaphase, KIF22 contributes to chromosome congression and alignment by generating polar ejection forces (*Brouhard and Hunt, 2005*; *Levesque and Compton, 2001*; *Stumpff et al., 2012*; *Wandke et al., 2012*). In cells treated with monastrol to inhibit Eg5/KIF11 and generate monopolar spindles, polar ejection forces push chromosomes away from a single central spindle pole (*Levesque and Compton, 2001*; *Figure 3A*). A loss of KIF22 function causes chromosomes to collapse in toward the pole in this system (*Levesque and Compton, 2001*; *Figure 3A*). To determine whether overexpression of KIF22-GFP with pathogenic mutations has a dominant effect on polar ejection force generation, wild-type or mutant KIF22-GFP-expressing HeLa-Kyoto cells were treated with monastrol to induce mitotic arrest with monopolar spindles. Relative polar ejection forces were compared by measuring the distance from the spindle pole to the maximum DAPI signal (*Figure 3A*). Expression of mutant motor did not reduce polar ejection forces (*Figure 3B and C*). Rather, expression of KIF22-GFP R149L and R149Q significantly increased the distance from the pole to the maximum DAPI signal (R149L 4.6±0.13 µm, R149Q 4.3±0.11 µm, and GFP control 3.7±0.04 µm, mean ± SEM), indicating higher levels of polar ejection forces in these cells.

The same assay was used to test whether mutant KIF22 could rescue polar ejection force generation in cells depleted of endogenous KIF22. In control cells expressing GFP, depletion of endogenous KIF22 resulted in the collapse of chromosomes toward the pole (*Figure 3B*), and the distance from the pole to the maximum DAPI signal was reduced to 1.6±0.11 µm, indicating a loss of polar ejection forces (*Figure 3D*). This reduction was not observed in cells expressing wild-type or mutant KIF22-GFP, demonstrating that KIF22-GFP with pathogenic mutations is capable of generating polar ejection forces (*Figure 3B and D*). In cells transfected with control siRNA and cells depleted of endogenous KIF22, polar ejection force levels did not depend on KIF22-GFP expression levels (*Figure 3E and F*).

Taken together, the localization of mutant KIF22 and the ability of mutant KIF22 to generate polar ejection forces indicate that pathogenic mutations P148L, P148S, R149L, R149Q, and V475G do not result in a loss of KIF22 function during early mitosis.

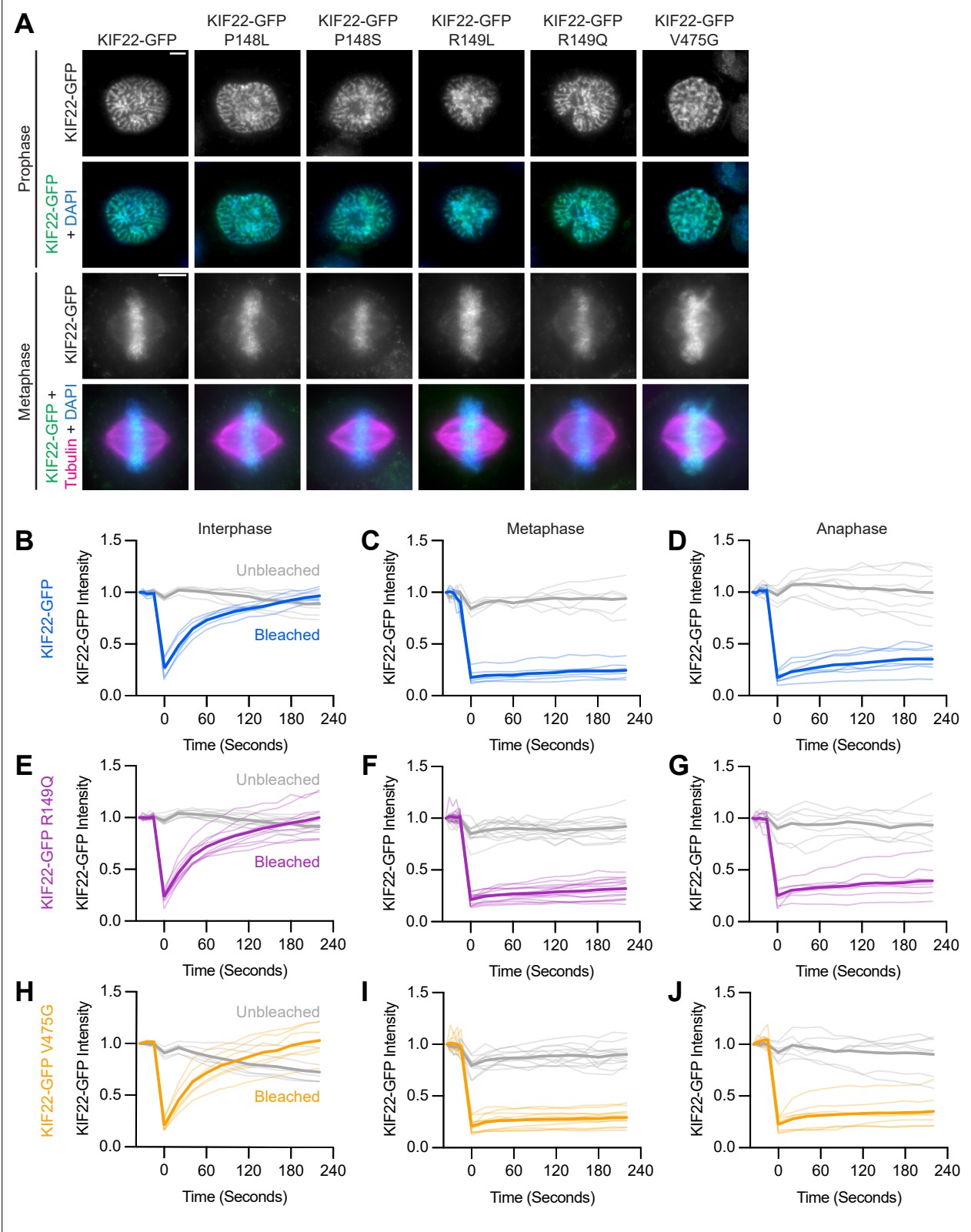

**Figure 2.** Pathogenic mutations in KIF22 do not disrupt the localization of the motor. (**A**) Immunofluorescence images of HeLa-Kyoto cells expressing KIF22-GFP constructs in prophase (top two rows) and metaphase (bottom two rows). KIF22-GFP was visualized using an anti-GFP antibody. Images are maximum intensity projections in z of five frames at the center of the spindle (metaphase cells) or maximum intensity projections in z of two frames (prophase cells). Fixed approximately 24 hr after treatment with doxycycline to induce expression. Scale bars 5 μm. (**B–J**) Fluorescence recovery after

*Figure 2 continued on next page*

Figure 2 continued

photobleaching (FRAP) of KIF22-GFP (**B–D**), KIF22-GFP R149Q (**E–G**), and KIF22-GFP V475G (**H–J**) in interphase nuclei (**B, E, H**) or on metaphase (**C, F, I**) or anaphase (**D, G, J**) chromosomes. Bleaching occurred at time 0. Thin lines are traces from individual cells and thick lines represent means. Intensity values are normalized to the KIF22-GFP intensity in the first imaged frame before bleaching. Interphase measurements (**B, E, H**) obtained from six KIF22-GFP cells from four experiments, nine KIF22-GFP R149Q cells from five experiments, and six KIF22-GFP V475G cells from four experiments. Metaphase measurements (**C, F, I**) obtained from 6 KIF22-GFP cells from four experiments, 14 KIF22-GFP R149Q cells from five experiments, and 12 KIF22-GFP V475G cells from four experiments. Anaphase measurements (**D, G, J**) obtained from eight KIF22-GFP cells from four experiments, seven KIF22-GFP R149Q cells from five experiments, and seven KIF22-GFP V475G cells from three experiments. See *Figure 2—source data 1*.

The online version of this article includes the following source data and figure supplement(s) for figure 2:

**Source data 1.** Fluorescence recovery after photobleaching (FRAP).

**Figure supplement 1.** HeLa-Kyoto and RPE-1 stable cell lines express mutant KIF22.

**Figure supplement 1—source data 1.** HeLa-Kyoto and RPE-1 cell lines.

**Figure supplement 2.** Pathogenic mutations in KIF22 do not disrupt the localization of the motor in RPE-1 cells.

## KIF22 mutations disrupt anaphase chromosome segregation

While pathogenic mutations did not disrupt the function of KIF22 in prometa- or metaphase, HeLa-Kyoto cells expressing mutant KIF22-GFP exhibited defects in anaphase chromosome segregation. In these cells, chromosomes did not move persistently toward the spindle poles. Instead, chromosomes began to segregate, but then reversed direction and moved back toward the center of the spindle or remained in the center of the spindle until decondensation (*Figure 4A*; *Video 2*). This phenotype was dominant and occurred in the presence of endogenous KIF22. Recongression was quantified by measuring the distance between separating chromosome masses as anaphase progressed. In cells expressing wild-type KIF22-GFP, this value increases steadily and then plateaus. Expression of mutant KIF22-GFP causes the distance between chromosome masses to increase, then decrease as chromosomes recongress, and then increase again as segregation continues (*Figure 4B*). Recongression reduces the distance between chromosome masses 7 min after anaphase onset in cells expressing KIF22-GFP with pathogenic mutations (median distance 2.0–7.2 µm) compared to cells expressing wild-type KIF22-GFP (median distance 12.9 µm) (*Figure 4C*). Defects in anaphase chromosome segregation were also observed in RPE-1 cells expressing KIF22-GFP R149Q or V475G (*Figure 4—figure supplement 1D-F*; *Video 3*). This gain of function phenotype is consistent with a lack of KIF22 inactivation in anaphase, resulting in a failure to suspend polar ejection force generation.

If recongression is the result of increased KIF22 activity in anaphase, we would predict that increased levels of KIF22-GFP expression would cause more severe anaphase chromosome segregation defects. Indeed, plotting the distance between chromosome masses 7 min after anaphase onset against mean GFP intensity for each HeLa-Kyoto cell demonstrated that these two values were correlated (Spearman correlation coefficient –0.6246, one-tailed p-value<0.0001) (*Figure 4—figure supplement 1A*). Considering only cells expressing lower levels of KIF22-GFP (mean background subtracted intensity<100 arbitrary units) emphasized the differences in the distance between chromosome masses

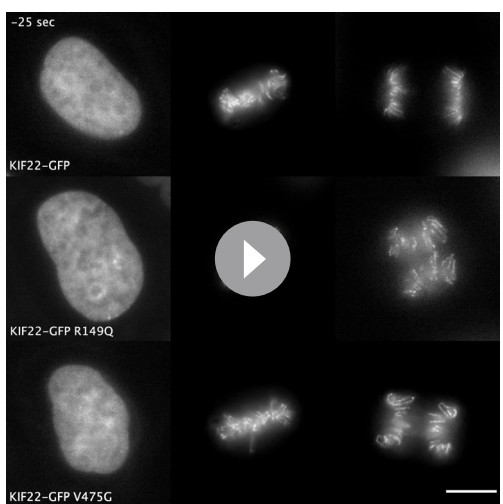

**Video 1.** Fluorescence recovery after photobleaching (FRAP) of KIF22-GFP. Fluorescence recovery after photobleaching in HeLa-Kyoto cells expressing KIF22-GFP (top), KIF22-GFP R149Q (middle), or KIF22-GFP V475G (bottom). Cells represent interphase (left), metaphase (middle), or anaphase (right). Bleaching occurred at time zero. Scale bar 10 µm. Cells were imaged at 5 second intervals for 25 seconds before bleaching, photobleached, and imaged at 20 second intervals for 10 minutes after bleaching. Playback at 10 frames per second.
https://elifesciences.org/articles/78653/figures#video1

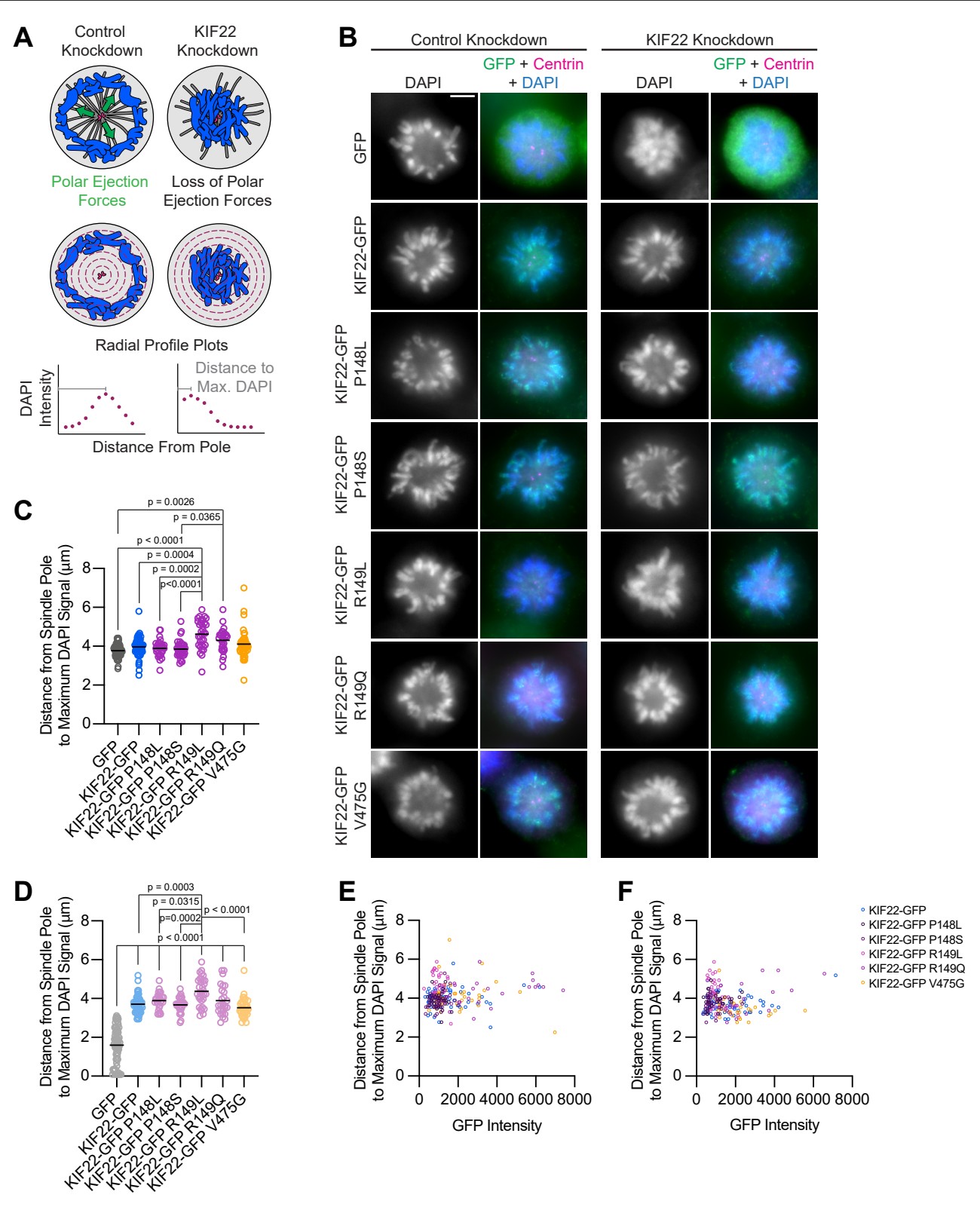

**Figure 3.** Pathogenic mutations in KIF22 do not reduce polar ejection forces. (**A**) Schematic of changes in chromosome positions resulting from loss of polar ejection forces. In cells with monopolar spindles, both spindle poles (magenta) are positioned together and chromosomes (blue) are pushed toward the cell periphery by polar ejection forces (green) (left). In cells depleted of KIF22, polar ejection forces are reduced and chromosomes collapse in toward the center of the cell (right). Relative polar ejection forces were quantified using radial profile plots to measure the distance from the spindle

*Figure 3 continued on next page*

*Figure 3 continued*

pole to the maximum DAPI signal intensity. (**B**) Immunofluorescence images of monopolar HeLa-Kyoto cells. KIF22-GFP was visualized using an anti-GFP antibody. Fixed approximately 2–3 hr after treatment with monastrol and 24 hr after siRNA transfection and treatment with doxycycline to induce expression. Scale bar 5 μm. Images are representative of three or more experiments. (**C**) Distance from the spindle pole to the maximum DAPI signal, a measure of relative polar ejection force level, in cells transfected with control siRNA. Fifty-nine GFP cells from seven experiments, 69 KIF22-GFP cells from six experiments, 31 KIF22-GFP P148L cells from three experiments, 37 KIF22-GFP P148S cells from three experiments, 33 KIF22-GFP R149L cells from three experiments, 28 KIF22-GFP R149Q cells from three experiments, and 45 KIF22-GFP V475G cells from three experiments. (**D**) Distance from the spindle pole to the maximum DAPI signal in cells transfected with KIF22 siRNA. Seventy-five GFP cells from seven experiments, 57 KIF22-GFP from six experiments, 28 KIF22-GFP P148L cells from three experiments, 30 KIF22-GFP P148S cells from three experiments, 33 KIF22-GFP R149L cells from three experiments, 26 KIF22-GFP R149Q cells from three experiments, and 34 KIF22-GFP V475G cells. For (**C, D**), bars indicate means. P values from Brown-Forsythe and Welch ANOVA with Dunnett's T3 multiple comparisons test. P values are greater than 0.05 for comparisons without a marked p value. (**E, F**) Background-subtracted GFP intensity plotted against the distance from the spindle pole to the maximum DAPI signal to assess dependence of polar ejection force generation on expression levels in cells transfected with control siRNA (**E**) (Pearson correlation coefficient 0.105, two-tailed p value 0.1031) or KIF22 siRNA (**F**) (Pearson correlation coefficient –0.005, two-tailed p value 0.9427). See *Figure 3—source data 1*.

The online version of this article includes the following source data for figure 3:

**Source data 1.** Polar ejection forces.

as anaphase progressed between cells expressing wild-type and mutant motor (*Figure 4—figure supplement 1B-C*).

In a subset of HeLa-Kyoto cells, expression of KIF22-GFP with pathogenic mutations caused cytokinesis failure (*Figure 4D*; *Video 4*). This result is consistent with the published observation that causing chromosome recongression by preventing cyclin B1 degradation can result in cytokinesis failure (*Wolf et al., 2006*). In cells expressing KIF22-GFP with pathogenic mutations, cleavage furrow ingression began, but did not complete, resulting in a single daughter cell. The percentage of cells failing to complete cytokinesis was approximately tenfold higher in cells expressing mutant KIF22-GFP (R149Q 36%, V475G 25%) than in cells expressing wild-type KIF22-GFP (3%). Additionally, the distance between chromosome masses at the time of cleavage furrow ingression was reduced in cells expressing KIF22-GFP R149Q or V475G, suggesting that the position of the chromosome masses may be physically obstructing cytokinesis (*Figure 4E*). Consistent with this hypothesis, cells that failed to complete cytokinesis tended to have lower distances between chromosome masses than the distances measured in cells in which cytokinesis completed despite the expression of mutant KIF22-GFP (*Figure 4E*).

## Mutations disrupt the separation of the spindle poles in anaphase

Anaphase chromosome segregation requires both that chromosome arms and centromeres move toward the spindle poles (anaphase A) (*Asbury, 2017*) and that the spindle poles move away from one another (anaphase B) (*Ris, 1949*). To test whether the activity of mutant KIF22 in anaphase affects one or both of these processes, anaphase was imaged in HeLa-Kyoto cells expressing fluorescent markers for the poles (pericentrin-RFP) and centromeres (CENPB-mCh) (*Figure 5A*). The reduced distance between separating chromosome masses seen in these cells (*Figure 5B and C*) was compared to the distances between the centromeres (*Figure 5D and E*) and the distances between the poles (*Figure 5F and G*) as anaphase progressed. The distances between all three structures showed the same trend: in cells expressing wild-type KIF22-GFP, the distance between chromosome masses, between centromeres, and between the spindle poles increased throughout the measured time interval in anaphase. Pathogenic mutations altered the movements of all three structures (*Figure 5B, D and F*; *Video 5*). The distance between chromosome masses, between centromeres, and between the spindle poles 10 min after anaphase onset was significantly reduced in cells expressing KIF22-GFP R149Q or KIF22-GFP V475G (*Figure 5C, E and G*). Comparing the distance between chromosome masses and the spindle pole within each half spindle (*Figure 5H*) with the distance between centromeres and the spindle pole in the same half spindles (*Figure 5I*) demonstrated that expression of mutant KIF22 more potently reduced the segregation of chromosome arms than centromeres, consistent with continued generation of polar ejection forces in anaphase. This suggests that pathogenic mutations in KIF22 affect anaphase A by altering the movement of chromosome arms, but not the shortening of the k-fibers, and affect anaphase B by altering spindle pole separation.

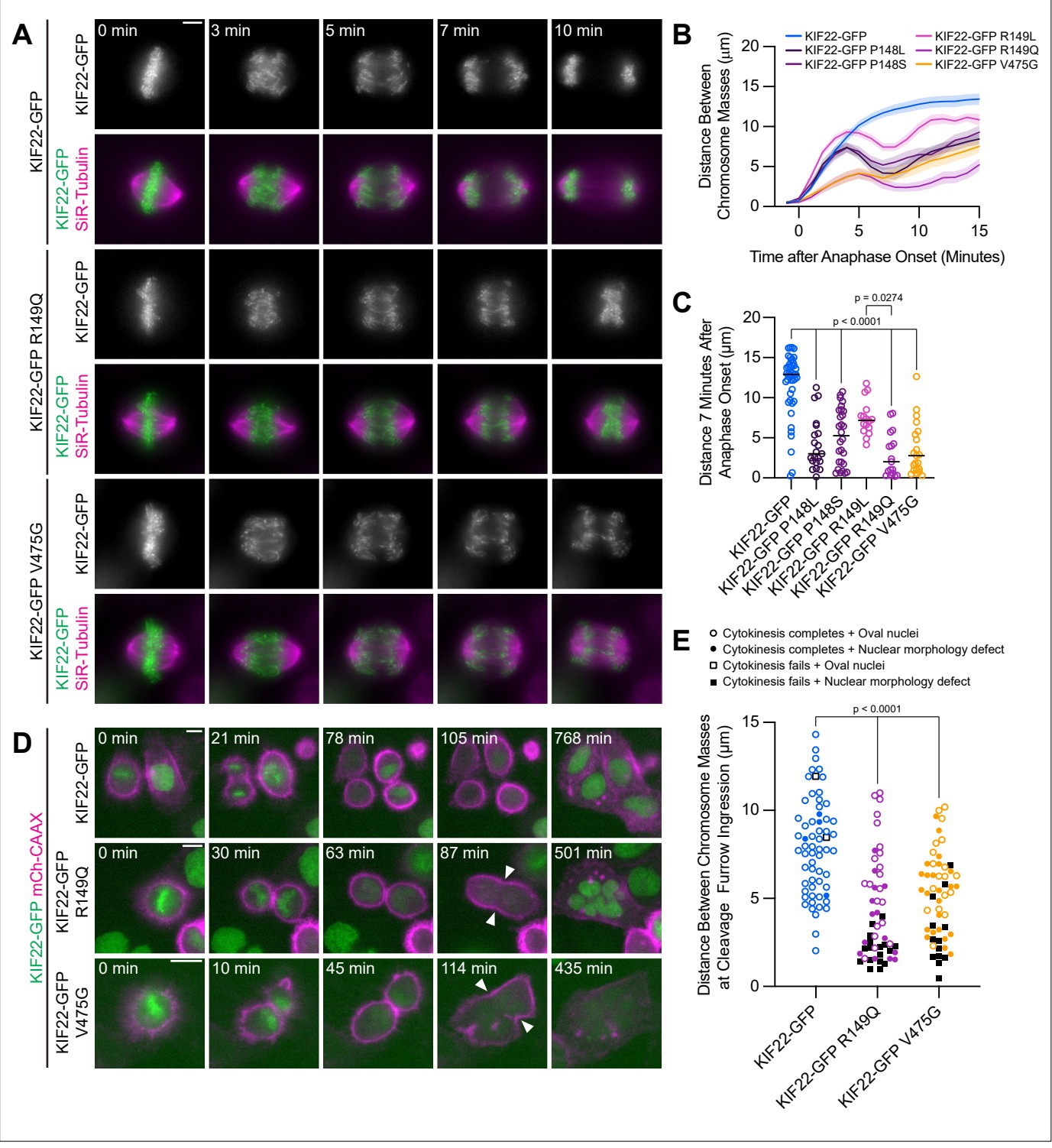

**Figure 4.** Pathogenic mutations in KIF22 disrupt anaphase chromosome segregation. (**A**) Time-lapse images of dividing HeLa-Kyoto cells expressing KIF22-GFP R149Q or KIF22-GFP V475G. Times indicate minutes after anaphase onset. Images are maximum intensity projections in z through the entirety of the spindle. Imaged approximately 18 hr after treatment with doxycycline to induce expression. Scale bar 5 µm. Images are representative of three or more experiments. (**B**) Distance between separating chromosome masses throughout anaphase in HeLa-Kyoto cells. Lines represent the mean and the shaded area denotes SEM. Forty-three KIF22-GFP cells from 10 experiments, 21 KIF22-GFP P148L cells from 6 experiments, 28 KIF22-GFP P148S cells from 7 experiments, 16 KIF22-GFP R149L cells from 6 experiments, 17 KIF22-GFP R149Q cells from 4 experiments, and 21 KIF22-GFP V475G cells from 21 experiments. (**C**) Distance between separating chromosome masses 7 min after anaphase onset. Bars indicate medians. P values from Kruskal-

*Figure 4 continued on next page*

*Figure 4 continued*

Wallis test. P values are greater than 0.05 for comparisons without a marked p value. Data represent the same cell populations presented in (**B**). (**D**) Time-lapse images of dividing HeLa-Kyoto cells expressing mCherry (mCh)-CAAX to visualize cell boundaries. Times indicate minutes after anaphase onset. Arrowheads indicate cytokinesis failure. Imaged approximately 8 hr after treatment with doxycycline to induce expression and 24–32 hr after transfection with mCh-CAAX. Scale bars 20 μm. Images are representative of three or more experiments. (**E**) Distance between chromosome masses at the time of cleavage furrow ingression. P values from Kruskal-Wallis test. P values are greater than 0.05 for comparisons without a marked p value. Sixty-two KIF22-GFP cells from 10 experiments, 52 KIF22-GFP R149Q cells from 9 experiments, and 55 KIF22-GFP V475G cells from 9 experiments. See *Figure 4—source data 1*.

The online version of this article includes the following source data and figure supplement(s) for figure 4:

**Source data 1.** Anaphase chromosome segregation and cytokinesis.

**Figure supplement 1.** Anaphase recongression defects are KIF22-GFP expression level dependent and disrupt chromosome segregation in RPE1 cells.

**Figure supplement 1—source data 1.** RPE-1 anaphase chromosome segregation.

## Division of cells expressing KIF22 with pathogenic mutations results in daughter cells with abnormally shaped nuclei

To understand the consequences of the observed defects in anaphase chromosome segregation, we examined the daughter cells produced by the division of cells expressing KIF22-GFP with pathogenic mutations. In these cells, the nuclei are lobed and fragmented (*Figure 6A*). The percentage of divisions resulting in nuclear morphology defects was approximately tenfold higher than in control cells (KIF22-GFP 6%, KIF22-GFP R149Q 64%, and KIF22-GFP V475G 68%) when live divisions were observed (*Figure 4E*). To further quantify this phenotype, the solidity of fixed cell nuclei (the ratio of the area of each nucleus to the area of the convex shape that would enclose it) was measured. A perfectly oval nucleus would have a solidity value of 1. Solidity values were reduced in cells expressing KIF22-GFP with pathogenic mutations (*Figure 6B*), indicating that these cells had more irregularly shaped nuclei. This reduction in solidity was dominant and occurred both in the presence of endogenous KIF22 and when endogenous KIF22 was depleted via siRNA knockdown. Using the fifth percentile solidity of control cells (control knockdown, GFP expression) as a cutoff, 44–63% of cells expressing mutant KIF22-GFP had abnormally shaped nuclei 24 hr after treatment with doxycycline to induce expression of KIF22-GFP (*Figure 6C*). Expression of wild-type KIF22-GFP also resulted in a small increase in the percentage of cells with abnormally shaped nuclei (12%). This percentage was reduced when endogenous KIF22 was depleted (7%), consistent with nuclear morphology defects resulting from an increase in KIF22 activity.

Expression of KIF22-GFP with pathogenic mutations also caused abnormally shaped nuclei in RPE-1 cells (*Figure 6—figure supplement 1A*). The solidity of nuclei in cells expressing mutant KIF22-GFP was reduced (*Figure 6—figure supplement 1B*), and 40–49% of RPE-1 cells expressing mutant KIF22-GFP had abnormally shaped nuclei, again defined as a solidity value less than the fifth percentile of control cells (*Figure 4C*). In RPE-1 cells, expression of wild-type KIF22-GFP resulted in a higher percentage of cells with abnormally shaped nuclei (18% in control knockdown cells, 15% with KIF22 knockdown) than was seen in HeLa-Kyoto

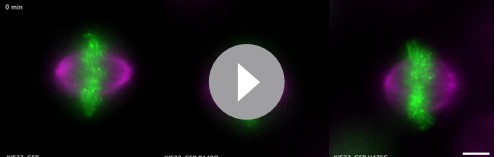

**Video 2.** Anaphase in HeLa-Kyoto cells. Anaphase chromosome segregation in HeLa-Kyoto cells expressing KIF22-GFP (left), KIF22-GFP R149Q (middle), or KIF22-GFP V475G (right). Magenta: SiR-Tubulin, green: KIF22-GFP. Times indicate minutes after anaphase onset. Scale bar 5 μm. Cells were imaged at 1 minute intervals. Playback at 10 frames per second (600 X real time).

https://elifesciences.org/articles/78653/figures#video2

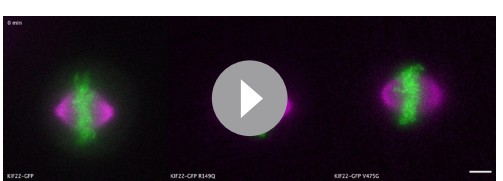

**Video 3.** Anaphase in RPE-1 cells. Anaphase chromosome segregation in RPE-1 cells expressing KIF22-GFP (left), KIF22-GFP R149Q (middle), or KIF22-GFP V475G (right). Magenta: SiR-Tubulin, green: KIF22-GFP. Times indicate minutes after anaphase onset. Scale bar 5 μm. Cells were imaged at 1 minute intervals. Playback at 10 frames per second (600 X real time).

https://elifesciences.org/articles/78653/figures#video3

**Video 4.** Cytokinesis and cytokinesis failure. Mitosis and cytokinesis in HeLa-Kyoto cells expressing KIF22-GFP (left), KIF22-GFP R149Q (middle), or KIF22-GFP V475G (right) (all KIF22-GFP represented in green) and mCh-CAAX (magenta). Scale bar 10 μm. Cells were imaged at 3 minute intervals. Playback at 25 frames per second (4500 X real time).

https://elifesciences.org/articles/78653/figures#video4

cells. This may be a result of the higher expression level of KIF22-GFP in the RPE-1 inducible cell lines (*Figure 2—figure supplement 1I,K*).

To determine whether these nuclear morphology defects depended on the ability of KIF22 to generate forces within the mitotic spindle, cells were treated with nocodazole to depolymerize microtubules and reversine to silence the spindle assembly checkpoint, allowing cells to enter and exit mitosis without assembling a spindle or segregating chromosomes (*Samwer et al., 2017*; *Serra-Marques et al., 2020*; *Figure 6D*). The solidity of nuclei was measured before chromosomes condensed (*Figure 6E*) and after mitotic exit (*Figure 6F*). At both time points, there was no difference in nuclear shape between control cells and cells expressing KIF22-GFP with pathogenic mutations, indicating that the effects of mutations on nuclear structure are spindle-dependent.

The effect of nuclear morphology defects on daughter cell fitness may partially depend on whether the nuclear envelopes of abnormally shaped nuclei are intact. The expression of mCherry (mCh) with a nuclear localization signal (NLS) indicated that even highly lobed and fragmented nuclei in cells expressing mutant KIF22-GFP are capable of retaining nuclear-localized proteins (*Figure 6G*). This suggests that the nuclear envelopes of these abnormally shaped nuclei are still intact enough to function as a permeability barrier (*Hatch et al., 2013*).

## Proliferation is reduced in cells expressing KIF22 with pathogenic mutations

If defects in anaphase chromosome segregation and nuclear morphology affect cellular function, they may impact the ability of cells to proliferate. To test this, HeLa-Kyoto cells expressing KIF22-GFP with pathogenic mutations were imaged over 96 hr to count the numbers of cells over time (*Figure 7A*). The growth rates of cells expressing mutant KIF22 were reduced (*Figure 7B*). After 96 hr, the fold change in cell number was reduced by approximately 30% for cells expressing KIF22-GFP with pathogenic mutations (GFP control median 5.3, KIF22-GFP R149Q 3.7, and KIF22-GFP V475G 3.8) (*Figure 7C*).

To consider what might be limiting the proliferation rate of cells expressing mutant KIF22-GFP, predictions for proliferation rate based on the observed rates of nuclear morphology defects and cytokinesis failure were calculated. For these purposes, only data from the first 48 hr of the proliferation assay were used, as cell growth rates plateaued after this time point. The doubling time of control HeLa-Kyoto cells expressing GFP was calculated to be 20.72 hr in these experiments, which is consistent with published data (*Liu et al., 2018*). Using this doubling rate, assuming exponential growth, and assuming every cell divides, the normalized cell count at 48 hr (normalized to a starting cell count of 1) was predicted to be 4.98. This is close to the experimental 48 hr cell count for control cells (4.60), and higher than the experimental 48 hr cell count for cells expressing KIF22-GFP R149Q (3.13) or V475G (3.60), as these cell lines have reduced proliferation (*Figure 7B*, square). If one assumed that cells with abnormally shaped nuclei stop dividing, given that approximately 60% of mutant KIF22-GFP cell divisions result in abnormally shaped nuclei (*Figure 4E*), the predicted cell count at 48 hr would be 2.18 (*Figure 7B*, triangle). This is lower than the experimental cell count for cells expressing mutant KIF22-GFP, suggesting that cells with abnormally shaped nuclei must be capable of additional divisions. If, instead, one assumed that only cells that fail cytokinesis (30% of cells; *Figure 4E*) stop dividing, the predicted cell count would be 3.42 (*Figure 7B*, diamond). This value is consistent with the experimental 48 hr cell count for cells expressing KIF22-GFP with pathogenic mutations (3.13–3.60), suggesting the rate of cytokinesis failure may limit the rate of proliferation in these cells. Consistent with this possibility, an increased number of large cells that may have failed cytokinesis are visible in proliferation assay images at 72 hr (*Figure 7A*). However, we note that both nuclear morphology defects and cytokinesis failure may contribute to the measured reduction in proliferation.

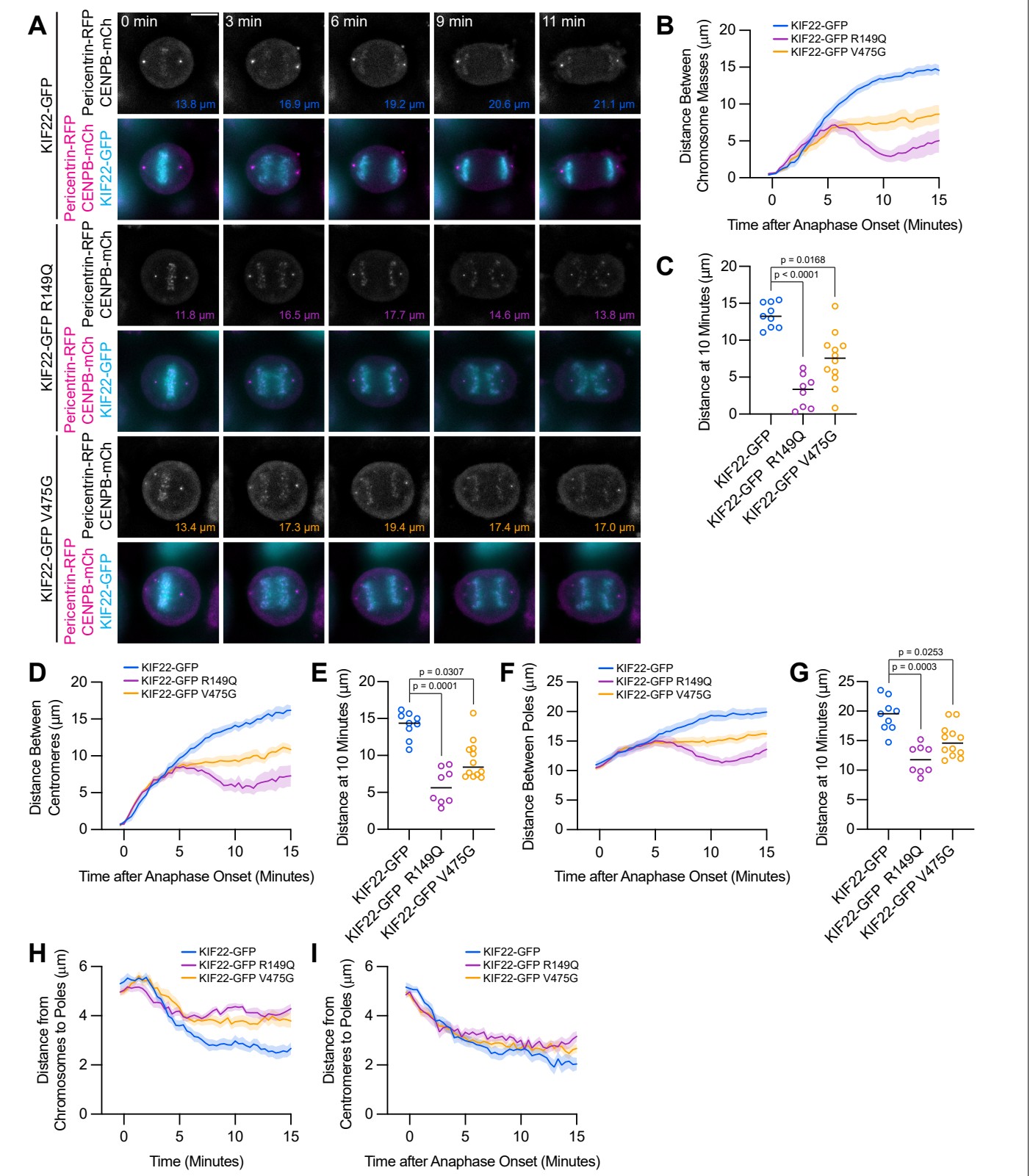

**Figure 5.** Mutations disrupt the separation of spindle poles in anaphase. (**A**) Time-lapse images of dividing HeLa-Kyoto cells expressing pericentrin-RFP to mark the spindle poles and CENPB-mCh to mark centromeres. Times indicate minutes after anaphase onset. Colored distances in the bottom right of each grayscale image indicate the distance between the spindle poles in the image. Images are maximum intensity projections in z through the entirety of the spindle. Imaged approximately 24 hr after transfection and 12–18 hr after treatment with doxycycline to induce expression. Images depicting

*Figure 5 continued on next page*

*Figure 5 continued*

pericentrin-RFP and CENPB-mCh signal were background subtracted by duplicating each frame, applying a gaussian blur (Sigma-Aldrich 30 pixels), and subtracting this blurred image from the original. Scale bar 10 µm. Images are representative of three or more experiments. (**B**) Distance between separating chromosome masses throughout anaphase in HeLa-Kyoto cells. Lines represent the mean and the shaded area denotes SEM. (**C**) Distance between separating chromosome masses 10 min after anaphase onset in HeLa-Kyoto cells. Bars indicate medians. (**D**) Distance between centromeres (CENPB-mCh) throughout anaphase in HeLa-Kyoto cells. Lines represent the mean and the shaded area denotes SEM. (**E**) Distance between centromeres 10 min after anaphase onset in HeLa-Kyoto cells. Bars indicate medians. (**F**) Distance between spindle poles (pericentrin-RFP) throughout anaphase in HeLa-Kyoto cells. Lines represent the mean and the shaded area denotes SEM. (**G**) Distance between spindle poles 10 min after anaphase onset in HeLa-Kyoto cells. Bars indicate medians. Measurements from the same cells (9 KIF22-GFP cells from five experiments, 8 KIF22-GFP R149Q cells from four experiments, and 12 KIF22-GFP V475G cells from six experiments) are shown in (**B–G**). For (**C, E, and G**), p values from Kruskal-Wallis test. (**H**) Distance between chromosome masses and spindle poles throughout anaphase in HeLa-Kyoto cells. Lines represent the mean and the shaded area denotes SEM. (**I**) Distance between centromeres and spindle poles throughout anaphase in HeLa-Kyoto cells. Lines represent the mean and the shaded area denotes SEM. Measurements from the same cells (18 KIF22-GFP, 16 KIF22-GFP R149Q, and 24 KIF22-GFP V475G half-spindles) as in (**B–G**) are shown in (**H**) and (**I**). See *Figure 5—source data 1*.

The online version of this article includes the following source data for figure 5:

**Source data 1.** Spindle pole and centromere distances.

To test the prediction that cells with nuclear morphology defects are capable of division, KIF22-GFP expression was induced approximately 24 hr before imaging to generate a population of cells with abnormally shaped nuclei. Division of these cells was observed (*Figure 7D*), demonstrating that nuclear morphology defects do not prevent subsequent divisions. The percentage of cells that divided over the course of this experiment was not reduced in cells expressing KIF22-GFP with pathogenic mutations despite the abnormal nuclear morphology of cells in those populations (*Figure 7E*).

## Mimicking phosphorylation of T463 phenocopies pathogenic mutations

The phenotypes observed in cells expressing KIF22-GFP with pathogenic mutations suggest that mutations may prevent inactivation of KIF22 in anaphase, and that polar ejection forces in anaphase disrupt chromosome segregation. If this is the case, then preventing KIF22 inactivation would be predicted to phenocopy the pathogenic mutations. One mechanism by which KIF22 activity is controlled is phosphorylation of T463: phosphorylation of this tail residue is necessary for polar ejection force generation, and dephosphorylation at anaphase onset contributes to polar ejection force suppression (*Soeda et al., 2016*). Therefore, we generated HeLa-Kyoto inducible cell lines expressing KIF22-GFP with phosphomimetic (T463D) and phosphonull (T463A) mutations to test whether preventing KIF22 inactivation in anaphase by expressing the constitutively active T463D construct phenocopies the expression of KIF22-GFP with pathogenic mutations. When treated with doxycycline, these cells expressed phosphomimetic and phosphonull KIF22-GFP at levels comparable to those seen in cell lines expressing KIF22-GFP with pathogenic mutations, which was approximately two- to threefold higher than the level of expression of endogenous KIF22 (*Figure 8—figure supplement 1A-D*).

To assess the activity of KIF22-GFP T463D and T463A in HeLa cells, polar ejection force generation in monopolar spindles was measured (*Figure 8A*). In cells with endogenous KIF22 present, expression of KIF22-GFP T463D increased the distance from the spindle pole to the maximum DAPI signal (GFP control 3.7±0.07 µm, KIF22-GFP T463D 4.4±0.12, mean ± SEM), indicating increased polar ejection forces, consistent with phosphorylation of T463 activating KIF22 in prometaphase (*Soeda et al., 2016*; *Figure 8B*). Conversely, when endogenous KIF22 was depleted, expression of KIF22-GFP T463A was less able to rescue polar ejection force generation (distance from the spindle pole to the maximum DAPI signal 3.0±0.08 µm, mean ± SEM) than expression of wild-type KIF22-GFP (3.6±0.07 µm) or KIF22-GFP T463D (3.7±0.10 µm) (*Figure 8C*). Again, this is consistent with previous work demonstrating that KIF22 phosphorylation at T463 activates the motor for prometaphase

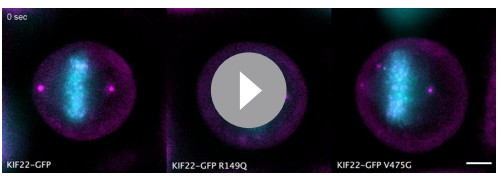

**Video 5.** Anaphase spindle pole separation. Anaphase in HeLa-Kyoto cells expressing pericentrin-RFP (magenta), CENPB-mCh (magenta), and KIF22-GFP (cyan). Times indicate seconds after anaphase onset. Scale bar 5 µm. Cells were imaged at 20 second intervals. Playback at 15 frames per second (300 X real time).

https://elifesciences.org/articles/78653/figures#video5

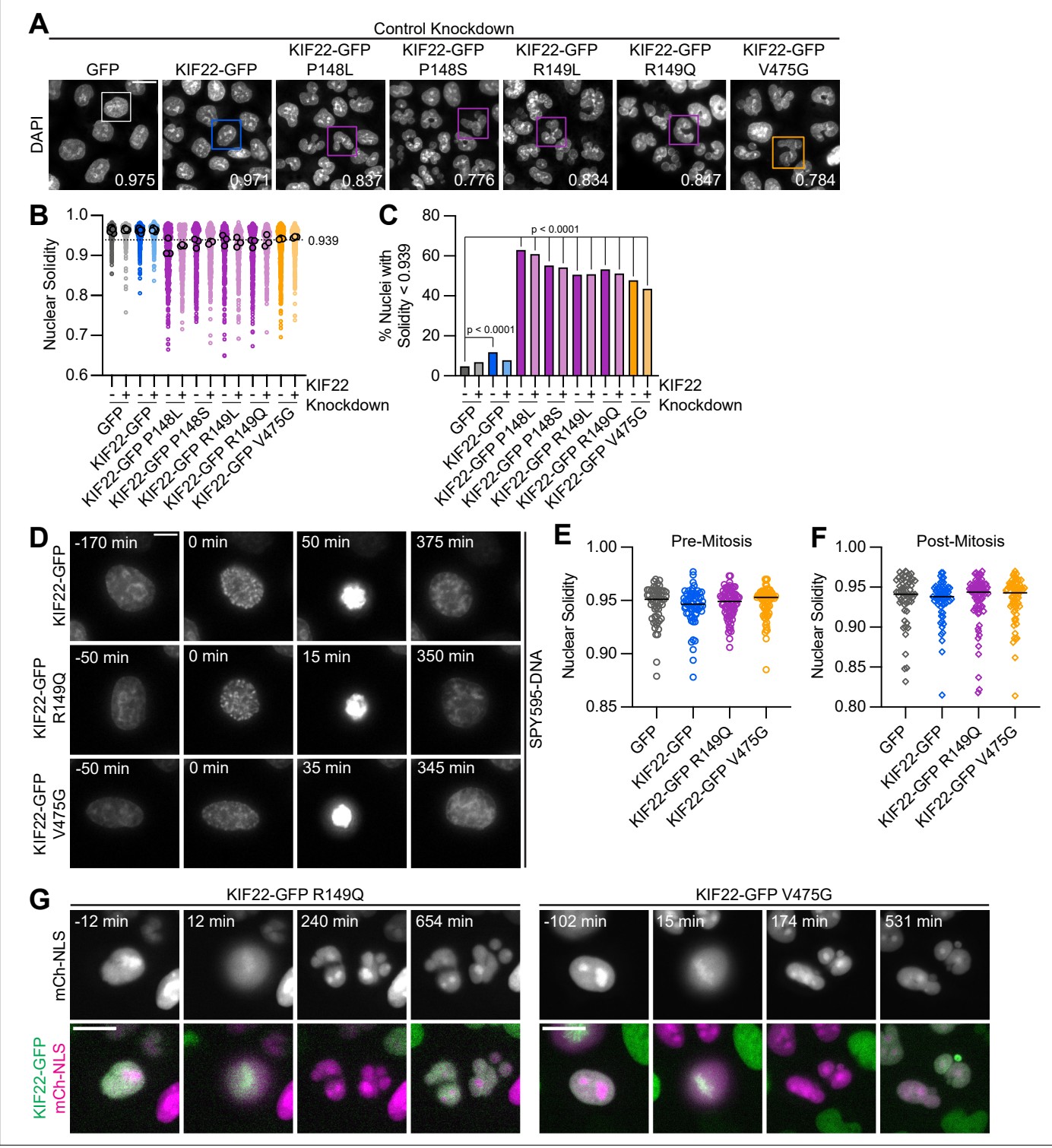

**Figure 6.** Division of cells expressing KIF22 with pathogenic mutations results in daughter cells with abnormally shaped nuclei. (**A**) DAPI stained nuclei of cells expressing KIF22 with pathogenic mutations. Values in the bottom right of each image indicate the solidity of the boxed nucleus. Fixed approximately 24 hr after treatment with doxycycline to induce expression. Scale bar 20 μm. Images are representative of three or more experiments. (**B**) Measured solidity of nuclei in HeLa-Kyoto cell lines. Small circles represent the solidity of individual nuclei, and large circles with black outlines indicate the median of each experiment. A dashed line marks a solidity value of 0.939, the fifth percentile of solidity for control cells transfected with control siRNA and expressing GFP. (**C**) Percentage of nuclei with abnormal shape, indicated by a solidity value less than 0.939, the fifth percentile of control (control knockdown, GFP expression) cell solidity. A chi-square test of all data produced a p-value<0.0001. Plotted p values are from pairwise

*Figure 6 continued on next page*

*Figure 6 continued*

post hoc chi-square tests comparing control (control knockdown, GFP expression) cells to each other condition. Applying the Bonferroni correction for multiple comparisons, a p value of less than 0.00385 was considered significant. P values are greater than 0.00385 for comparisons without a marked p value. Data in (**B**) and (**C**) represent 1045 GFP cells transfected with control siRNA, 849 GFP cells transfected with KIF22 siRNA, 994 KIF22-GFP cells transfected with control siRNA, 980 KIF22-GFP cells transfected with KIF22 siRNA, 472 KIF22-GFP P148L cells transfected with control siRNA, 442 KIF22-GFP P148L cells transfected with KIF22 siRNA, 382 KIF22-GFP P148S cells transfected with control siRNA, 411 KIF22-GFP P148S cells transfected with KIF22 siRNA, 336 KIF22-GFP R149L cells transfected with control siRNA, 376 KIF22-GFP R149L cells transfected with KIF22 siRNA, 466 KIF22-GFP R149Q cells transfected with control siRNA, 359 KIF22-GFP R149Q cells transfected with KIF22 siRNA, 605 KIF22-GFP V475G cells transfected with control siRNA, and 386 KIF22-GFP V475G cells transfected with KIF22 siRNA. GFP and KIF22-GFP cells represent six experiments, data from all other cell lines represent three experiments. (**D**) Time-lapse images of HeLa-Kyoto cells treated with nocodazole and reversine and stained with SPY595-DNA to visualize chromosomes. Time indicates the number of minutes before or after chromosome condensation. Images are maximum intensity projections in z of two focal planes, one at the level of interphase nuclei and one at the level of mitotic chromosomes. Imaged approximately 8 hr after treatment with doxycycline to induce expression, 1.5–2 hr after treatment with SPY595-DNA, and 0.5–1 hr after treatment with nocodazole and reversine. Scale bar 10 µm. Images are representative of three or more experiments. (**E**) Nuclear solidity of HeLa-Kyoto cells treated with nocodazole and reversine. Measurements were made 15 min before chromosome condensation. (**F**) Nuclear solidity of HeLa-Kyoto cells treated with nocodazole and reversine. Measurements were made 100 min after chromosome decondensation. Data in (**E**) and (**F**) represent 56 GFP, 60 KIF22-GFP, 76 KIF22-GFP R149Q, and 67 KIF22-GFP V475G cells from three experiments per condition. For (**E**) and (**F**), bars indicate medians, and the Kruskal-Wallis test indicated no significant difference between groups. (**G**) Time-lapse images of HeLa-Kyoto cells expressing mCherry (mCh)-NLS to assess nuclear envelope integrity. Times indicate minutes before or after chromosome condensation. Imaged approximately 8 hr after treatment with doxycycline to induce expression and 24 hr after transfection with mCh-CAAX. Scale bar 20 µm. Images are representative of three or more experiments. See *Figure 6—source data 1*.

The online version of this article includes the following source data and figure supplement(s) for figure 6:

**Source data 1.** Nuclear morphology.

**Figure supplement 1.** Mutations cause abnormally shaped nuclei in RPE1 cells.

**Figure supplement 1—source data 1.** RPE-1 nuclear morphology.

polar ejection force generation (*Soeda et al., 2016*), although the reduction in polar ejection forces seen with KIF22-GFP T463A rescue is less severe in our system, possibly due to differences in cell type, level of depletion of endogenous KIF22, or the method used to quantify polar ejection forces.

In anaphase, expression of phosphomimetic KIF22-GFP T463D, but not phosphonull KIF22-GFP T463A, caused chromosome recongression (*Figure 8D and E*). The distance between chromosome masses at 7 min was reduced in cells expressing KIF22-GFP T463D (median 5.8 µm) compared to cells expressing wild-type KIF22-GFP (12.5 µm) or KIF22-GFP T463A (10.8 µm) (*Figure 8F*). As in cells expressing KIF22-GFP with pathogenic mutations, the severity of anaphase chromosome recongression, indicated by the distance between chromosome masses at 7 min, was dependent on GFP expression level (Spearman correlation coefficient –0.3964, one-tailed p value 0.0004) (*Figure 8—figure supplement 1E*). When only cells expressing lower levels of KIF22-GFP (mean background subtracted intensity<100 arbitrary units) were considered, the same effect (expression of KIF22-GFP T463D causes recongression) was still observed (*Figure 8—figure supplement 1F-G*). This recongression phenocopies the effect of pathogenic mutations on anaphase chromosome segregation, consistent with pathogenic mutations preventing anaphase inactivation of KIF22.

In addition to causing the same defects in anaphase chromosome segregation, expression of KIF22-GFP T463D also affects daughter cell nuclear morphology. Cells expressing KIF22-GFP T463D have lobed and fragmented nuclei (*Figure 8—figure supplement 1H*) and correspondingly reduced nuclear solidity measurements (*Figure 8G*). An increased percentage of cells expressing KIF22-GFP T463D in the presence of endogenous KIF22 (65%) or in cells depleted of endogenous KIF22 (72%) have abnormally shaped nuclei, as indicated by a solidity value below the fifth percentile of control cell nuclear solidity (*Figure 8H*).

Expression of KIF22-GFP T463A also resulted in a small increase in the percentage of abnormally shaped nuclei (26% in control or KIF22 knockdown conditions) (*Figure 8H*). Since expression of KIF22-GFP T463A does not cause anaphase recongression (*Figure 8E*), the level of compaction of the segregating chromosome masses was explored as a possible explanation for this modest increase in the percentage of cells with nuclear morphology defects. In KIF22 knockout mice, loss of KIF22 reduces chromosome compaction in anaphase, causing the formation of multinucleated cells (*Ohsugi et al., 2008*). The phosphonull T463A mutation reduces KIF22 activity and may therefore exhibit a KIF22 loss of function phenotype. Measurement of the widths of separating chromosome masses in anaphase (*Figure 8—figure supplement 1 I*) did demonstrate a modest broadening of the chromosome masses

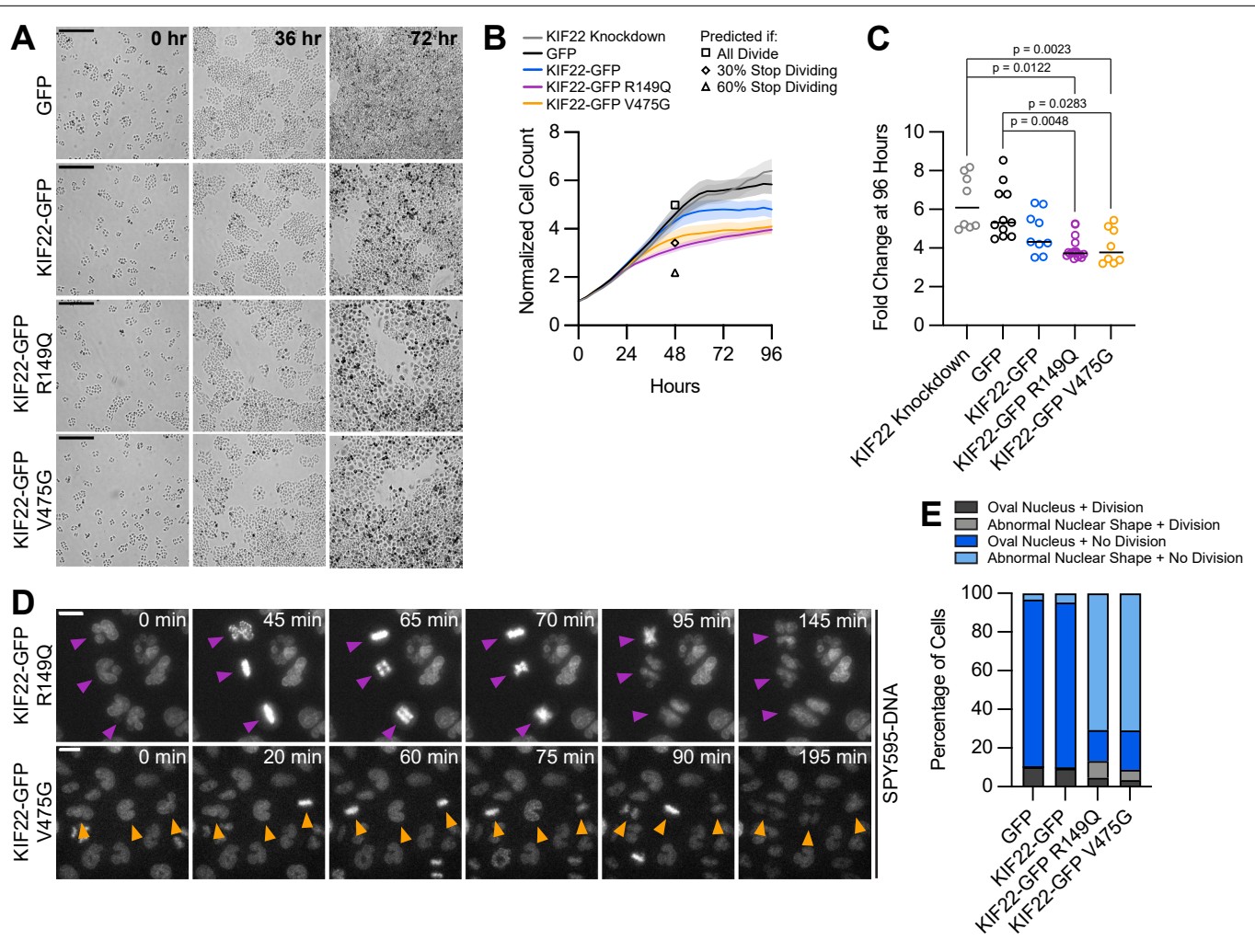

**Figure 7.** Proliferation is reduced in cells expressing KIF22 with pathogenic mutations. (**A**) Time-lapse bright field images of HeLa-Kyoto cells to assess proliferation rate. Scale bar 500 µm. Images are representative of three or more experiments. (**B**) Proliferation rates measured using automated bright field imaging. Lines represent the mean cell count, normalized to the number of cells at 0 hr, and the shaded area denotes SEM. Black outlined shapes indicate the predicted cell count for cell lines expressing pathogenic mutations at 48 hr if every cell doubled every 20.72 hr (the doubling time measured from 48 hr of control cell proliferation) (square), if the rate of cytokinesis failure limited proliferation and 30% of cells did not divide (diamond), and if the rate of nuclear morphology defects limited proliferation and 60% of cells did not divide (triangle). (**C**) Fold change of normalized cell counts after 96 hr. Bars indicate medians. P values from Kruskal-Wallis test. P values are greater than 0.05 for comparisons without a marked p value. Data in (**B**) and (**C**) represent 8 KIF22 knockdown, 11 GFP, 9 KIF22-GFP, 16 KIF22-GFP R149Q, and 8 KIF22-GFP V475G technical replicates from four experiments. (**D**) Time-lapse imaging of HeLa-Kyoto cells treated with doxycycline for 24 hr to induce expression of KIF22-GFP with pathogenic mutations and stained with SPY595-DNA. Arrowheads indicate cells with abnormally shaped nuclei that divide. Images are maximum intensity projections in z of two focal planes, one at the level of interphase nuclei and one at the level of mitotic chromosomes. Scale bars 20 µm. Images are representative of three or more experiments. (**E**) Nuclear morphology at the start of imaging (dark gray or blue, oval; light gray or blue; abnormal morphology) and outcome (gray, cell divides during the experiment; blue, the cell does not divide). The total number of dividing cells was compared between cell lines using the chi-square test (p<0.0001 across all conditions). Post hoc chi-square tests comparing all conditions to one another indicated that the proliferation rate of cells expressing KIF22-GFP R149Q is statistically different than that of cells expressing GFP (p=0.0025), KIF22-GFP (p=0.0003), or KIF22-GFP V475G (p<0.0001). Applying the Bonferroni correction for multiple comparisons, a p value of less than 0.008 was considered significant. P values are greater than 0.008 for all other comparisons. 2461 GFP, 2611 KIF22-GFP, 1890 KIF22-GFP R149Q, and 2346 KIF22-GFP V465G cells. See *Figure 7—source data 1*.

The online version of this article includes the following source data for figure 7:

**Source data 1.** Proliferation.

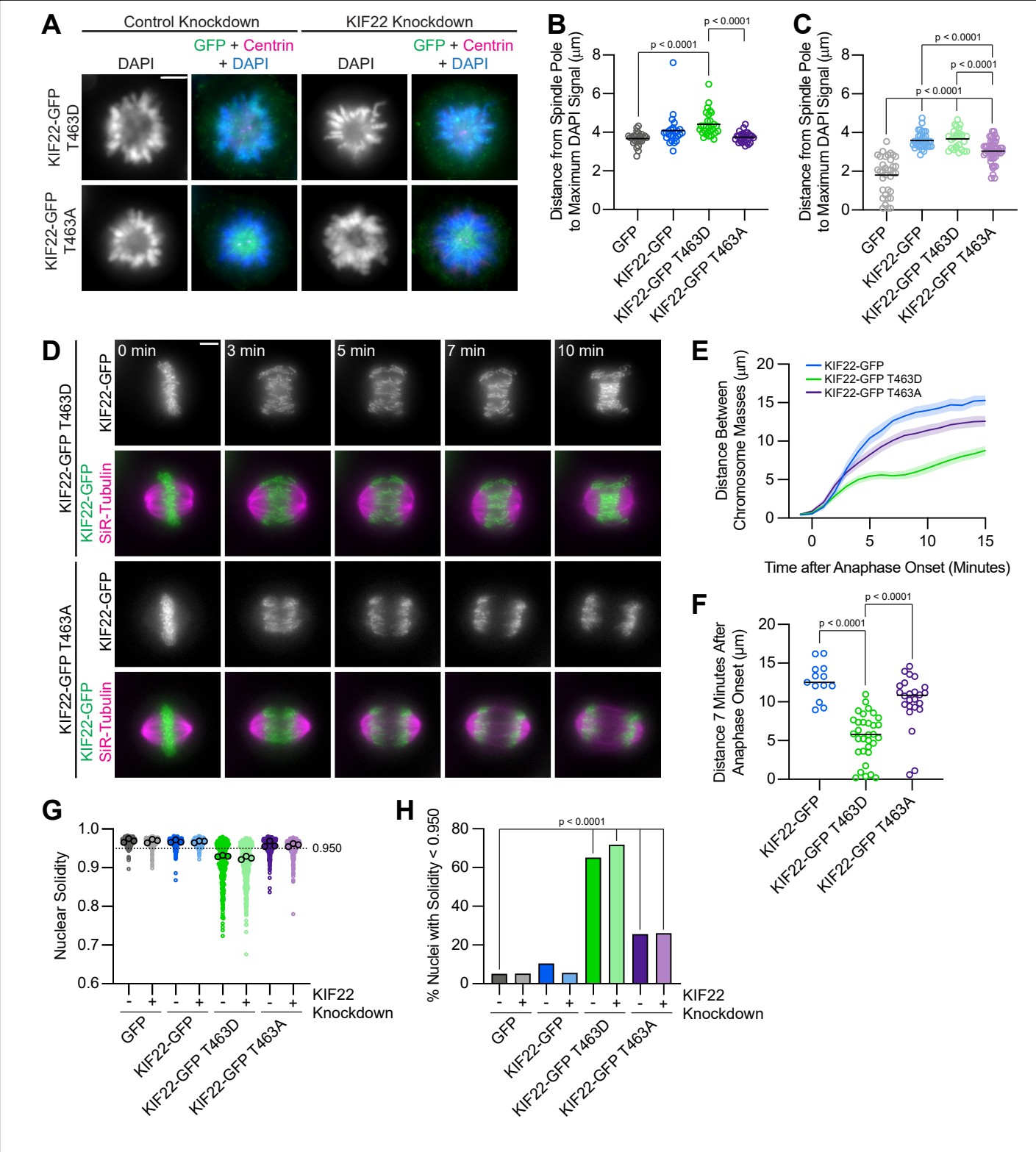

**Figure 8.** Phosphomimetic mutation of T463 phenocopies pathogenic mutations in KIF22. (**A**) Immunofluorescence images of monopolar HeLa-Kyoto cells. KIF22-GFP was visualized using an anti-GFP antibody. Fixed approximately 2–3 hr after treatment with monastrol and 24 hr after siRNA transfection and treatment with doxycycline to induce expression. Scale bar 5 μm. Images are representative of three or more experiments. (**B**) Distance from the spindle pole to the maximum DAPI signal, a measure of relative polar ejection force level, between HeLa-Kyoto cell lines expressing KIF22-GFP with phosphomimetic and phosphonull mutations at T463. Twenty-six GFP cells from three experiments, 26 KIF22-GFP cells from three experiments,

*Figure 8 continued on next page*

*Figure 8 continued*

29 KIF22-GFP T463D cells from three experiments, and 29 KIF22-GFP T463A cells from three experiments. (**C**) Distance from the spindle pole to the maximum DAPI signal in cells depleted of endogenous KIF22 and expressing KIF22-GFP with phosphomimetic and phosphonull mutations at T463. Thirty-five GFP cells from four experiments, 36 KIF22-GFP cells from four experiments, 27 KIF22-GFP T463D cells from three experiments, and 47 KIF22-GFP T463A cells from four experiments. For (**B, C**), bars indicate means. P values from Brown-Forsythe and Welch ANOVA with Dunnett's T3 multiple comparisons test. P values are greater than 0.05 for comparisons without a marked p value. (**D**) Time-lapse images of dividing HeLa-Kyoto cells. Cells expressing KIF22-GFP T463D exhibit recongression of the chromosomes during anaphase. Times indicate minutes after anaphase onset. Images are maximum intensity projections in z through the entirety of the spindle. Imaged approximately 18 hr after treatment with doxycycline to induce expression. Scale bar 5 µm. Images are representative of three or more experiments. (**E**) Distance between separating chromosome masses throughout anaphase in HeLa-Kyoto cells. Lines represent the mean and the shaded area denotes SEM. Thirteen KIF22-GFP, 32 KIF22-GFP T463D, and 24 KIF22-GFP T463A cells from five experiments. (**F**) Distance between separating chromosome masses 7 min after anaphase onset. Bars indicate medians. P values from Kruskal-Wallis test. P values are greater than 0.05 for comparisons without a marked p value. Thirteen KIF22-GFP, 32 KIF22-GFP T463D, and 24 KIF22-GFP T463A cells from five experiments per condition. (**G**) Measured solidity of nuclei in HeLa-Kyoto cell lines. Small circles represent the solidity of individual nuclei, and large circles with black outlines indicate the median of each experiment. A dashed line marks a solidity value of 0.950, the fifth percentile of solidity for control cells transfected with control siRNA and expressing GFP. (**H**) Percentage of nuclei with abnormal shape, indicated by a solidity value less than 0.950, the fifth percentile of control (control knockdown, GFP expression) cell solidity. A chi-square test of all data produced a p-value<0.0001. Plotted p values are from pairwise post hoc chi-square tests comparing control (control knockdown, GFP expression) cells to each other condition. Applying the Bonferroni correction for multiple comparisons, a p value of less than 0.00714 was considered significant. P values are greater than 0.00714 for comparisons without a marked p value. Data in (**G**) and (**H**) represent 312 GFP cells transfected with control siRNA, 362 GFP cells transfected with KIF22 siRNA, 314 KIF22-GFP cells transfected with control siRNA, 320 KIF22-GFP cells transfected with KIF22 siRNA, 361 KIF22-GFP T463D cells transfected with control siRNA, 376 KIF22-GFP T463D cells transfected with KIF22 siRNA, 312 KIF22-GFP T463A cells transfected with control siRNA, and 376 KIF22-GFP T463A cells transfected with KIF22 siRNA from three experiments. See *Figure 8—source data 1*.

The online version of this article includes the following source data and figure supplement(s) for figure 8:

**Source data 1.** T463 phosphomutants.

**Figure supplement 1.** Cells expressing KIF22-GFP T463A have broader anaphase chromosome masses.

**Figure supplement 1—source data 1.** T463 KIF22 expression level.

in cells expressing KIF22-GFP T-463A (*Figure 8—figure supplement 1* J-K), which may contribute to the modest defects in nuclear morphology seen in these cells.

## Mimicking phosphorylation of T158 in the α2 helix phenocopies pathogenic mutations

The effect of mutations in the α2 helix on KIF22 function suggests the involvement of this region of the motor domain in KIF22 inactivation. If this was true, post-translational modification of α2 may contribute to the regulation of KIF22 activity, analogous to the regulation of KIF22 inactivation via the dephosphorylation of T463 in the tail. Phosphorylation of amino acids T134 in α2a (*Kettenbach et al., 2011*) and T158 in α2b (*Olsen et al., 2010*; *Rigbolt et al., 2011*) has been documented in phosphoproteomic studies. HeLa-Kyoto cells expressing KIF22-GFP with phosphomimetic and phosphonull mutations at T134 and T158 were generated to test whether either site may contribute to the regulation of KIF22 inactivation.

T134 is located in α2a, near the catalytic site of KIF22 (*Figure 9—figure supplement 1* A). Both phosphonull (T134A) and phosphomimetic (T134D) mutations at this site disrupted the localization of KIF22. KIF22-GFP T134D and T134A localized to spindle microtubules rather than to the chromosomes when expressed at levels comparable to or lower than those of wild-type KIF22-GFP (*Figure 9—figure supplement 1B-F*). Expression of KIF22-GFP T134D and KIF22-GFP T134A also resulted in the formation of multipolar spindles in a subset of cells (*Figure 9—figure supplement 1* G). These phenotypes are consistent with previous work that used T134N as a rigor mutation to test the necessity of KIF22 motor activity for spindle length maintenance (*Tokai-Nishizumi et al., 2005*). The phenotypes observed in cells expressing KIF22-GFP T134D or KIF22-GFP T134A are not the same as those observed in cells expressing KIF22-GFP T463D, suggesting that phosphoregulation of T134 is not involved in the inactivation of KIF22.

T158 is located in α2b, the same region of the α2 helix containing amino acids P148 and R149, which are mutated in patients with SEMDJL2 (*Figure 9A*). Localization of phosphomimetic (T158D) or phosphonull (T158A) mutant KIF22-GFP was not altered compared to wild-type motor, and KIF22-GFP T158D or T158A expression levels were comparable to levels measured in cells expressing wild-type KIF22-GFP or KIF22-GFP with pathogenic mutations (*Figure 9—figure supplement 2A-D*).

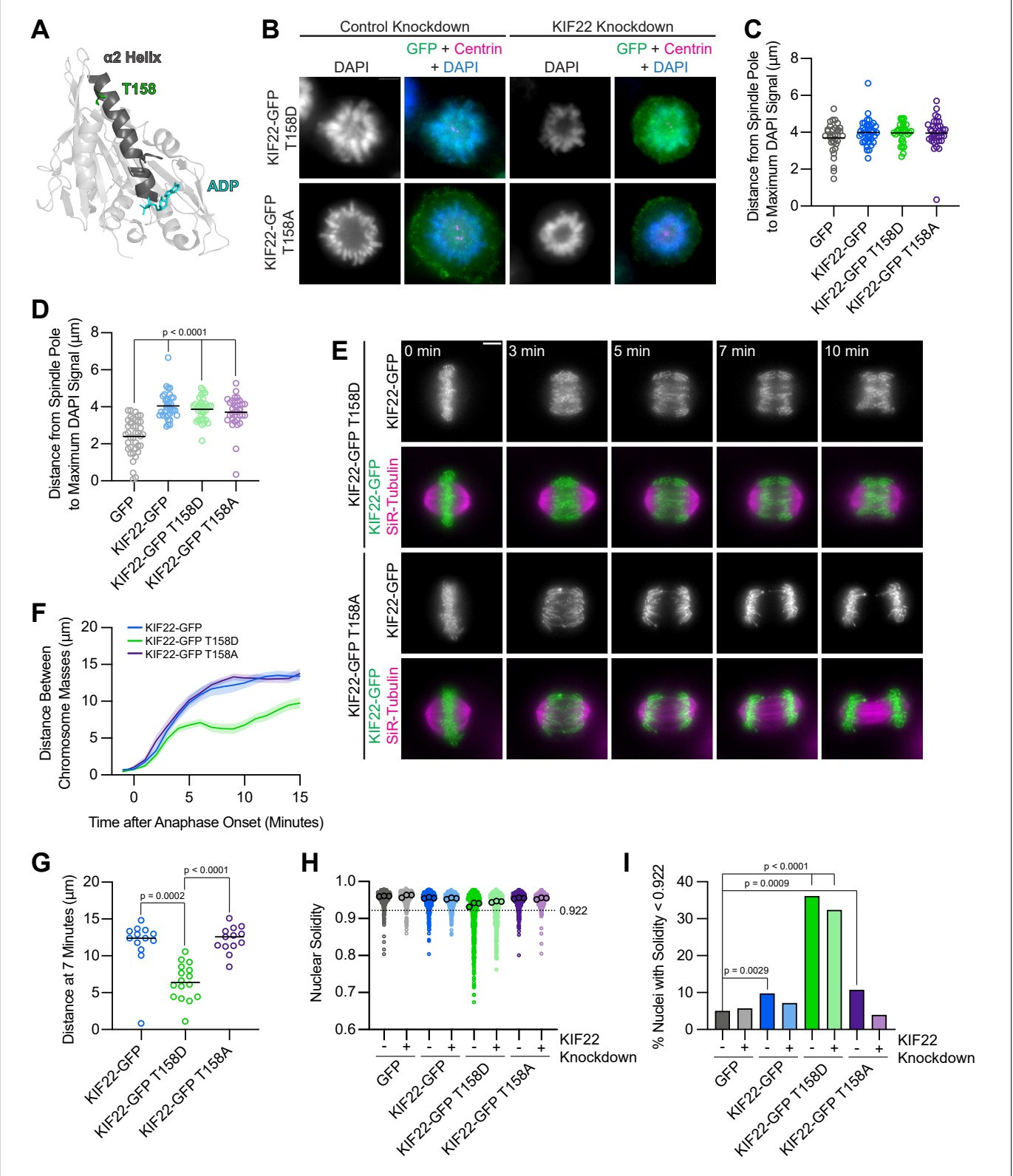

**Figure 9.** Mimicking phosphorylation of T158 in the motor domain affects KIF22 inactivation. (**A**) Location of amino acid T158 in the α2 helix of the KIF22 motor domain (PDB 6NJE). (**B**) Immunofluorescence images of monopolar HeLa-Kyoto cells. KIF22-GFP was visualized using an anti-GFP antibody. Fixed approximately 2–3 hr after treatment with monastrol and 24 hr after siRNA transfection and treatment with doxycycline to induce expression. Scale bar 5 µm. Images are representative of three or more experiments. (**C**) Distance from the spindle pole to the maximum DAPI signal, a measure of relative

*Figure 9 continued on next page*

*Figure 9 continued*

polar ejection force level, in HeLa-Kyoto cell lines expressing KIF22-GFP with phosphomimetic and phosphonull mutations at T158. Thirty-three GFP, 40 KIF22-GFP, 31 KIF22-GFP T158D, and 36 KIF22-GFP T158A cells from three experiments. (**D**) Distance from the spindle pole to the maximum DAPI signal in cells depleted of endogenous KIF22 and expressing KIF22-GFP with phosphomimetic and phosphonull mutations at T158. Thirty-nine GFP, 35 KIF22-GFP, 34 KIF22-GFP T158D, and 34 KIF22-GFP T158A cells from three experiments. For (**C, D**), bars indicate means. P values from Brown-Forsythe and Welch ANOVA with Dunnett's T3 multiple comparisons test. P values are greater than 0.05 for comparisons without a marked p value. (**E**) Time-lapse images of dividing HeLa-Kyoto cells. Cells expressing KIF22-GFP T158D exhibit recongression of the chromosomes during anaphase. Times indicate minutes after anaphase onset. Images are maximum intensity projections in z through the entirety of the spindle. Imaged approximately 18 hr after treatment with doxycycline to induce expression. Scale bar 5 µm. Images are representative of three or more experiments. (**F**) Distance between separating chromosome masses throughout anaphase in HeLa-Kyoto cells. Lines represent the mean and the shaded area denotes SEM. Thirteen KIF22-GFP, 16 KIF22-GFP T158D, and 13 KIF22-GFP T158A cells from five experiments. (**G**) Distance between separating chromosome masses 7 min after anaphase onset. Bars indicate medians. P values from Kruskal-Wallis test. P values are greater than 0.05 for comparisons without a marked p value. Thirteen KIF22-GFP, 16 KIF22-GFP T158D, and 13 KIF22-GFP T158A cells from five experiments. (**H**) Measured solidity of nuclei in HeLa-Kyoto cell lines. Small circles represent the solidity of individual nuclei, and large circles with black outlines indicate the median of each experiment. A dashed line marks a solidity value of 0.922, the fifth percentile of solidity for control cells transfected with control siRNA and expressing GFP. (**I**) Percentage of nuclei with abnormal shape, indicated by a solidity value less than 0.922, the fifth percentile of control (control knockdown, GFP expression) cell solidity. A chi-square test of all data produced a p-value<0.0001. Plotted p values are from pairwise post hoc chi-square tests comparing control (control knockdown, GFP expression) cells to each other condition. Applying the Bonferroni correction for multiple comparisons, a p value of less than 0.00714 was considered significant. P values are greater than 0.00714 for comparisons without a marked p value. Data in (**H**) and (**I**) represent 514 GFP control knockdown, 418 GFP KIF22 knockdown, 613 KIF22-GFP control knockdown, 584 KIF22-GFP KIF22 knockdown, 644 KIF22-GFP T158D control knockdown, 432 KIF22-GFP T158D KIF22 knockdown, 477 KIF22-GFP T158A control knockdown, and 427 KIF22-GFP T158A KIF22 knockdown cells from three experiments. See *Figure 9—source data 1*.

The online version of this article includes the following source data and figure supplement(s) for figure 9:

**Source data 1.** T158 phosphomutants.

**Figure supplement 1.** Mimicking phosphoregulation of T134 disrupts KIF22 localization.

**Figure supplement 1—source data 1.** T134 phosphomutants.

**Figure supplement 2.** Analysis of KIF22 expression level effects on chromosome recongression in cells expressing KIF22-GFP T158D and KIF22-GFP T158A.

**Figure supplement 2—source data 1.** T158 KIF22 expression level.

**Figure supplement 3.** KIF22 motor domain and tail fragments do not co-immunoprecipitate.

**Figure supplement 3—source data 1.** Western blot.

To assess the activity of KIF22-GFP T158D and KIF22-GFP T158A, relative polar ejection forces were measured in monopolar spindles (*Figure 9B*). In the presence of endogenous KIF22, expression of neither KIF22-GFP T158D nor KIF22-GFP T158A disrupted the generation of polar ejection forces (*Figure 9C*). In cells depleted of endogenous KIF22, expression of KIF22-GFP, KIF22-GFP T158D, or KIF22-GFP T158A was sufficient to rescue polar ejection force generation (*Figure 9D*), indicating that KIF22 with mutations at T158 is active in prometaphase and capable of generating polar ejection forces.

To test the effects of phosphomimetic and phosphonull mutations at T158 in anaphase, distances between separating chromosome masses in cells expressing KIF22-GFP, KIF22-GFP T158D, or KIF22-GFP T158A were measured. Expression of KIF22-GFP T158D caused chromosome recongression, while expression of KIF22-GFP T158A did not affect chromosome movements in anaphase (*Figure 9E and F*). The distance between separating chromosome masses 7 min after anaphase onset was reduced in cells expressing KIF22-GFP T158D (median 6.4 µm) compared to cells expressing KIF22-GFP (12.4 µm) or KIF22-GFP T158A (13.6 µm) (*Figure 9G*). Anaphase recongression was correlated with KIF22 expression levels (Spearman correlation coefficient –0.3647, one-tailed p value 0.0088) (*Figure 9—figure supplement 2E*), but when only cells with lower levels of KIF22-GFP expression (mean background subtracted intensity<100 arbitrary units) were considered the same trends in recongression were still observed (*Figure 9—figure supplement 2F, G*). Mimicking phosphorylation of T158 also affected daughter cell nuclear morphology. Nuclear solidity was reduced in cells expressing KIF22-GFP T158D (*Figure 9H*), and correspondingly the percentage of cells with abnormally shaped nuclei, designated as a solidity value lower than the fifth percentile solidity of control cells expressing GFP, was increased in cells expressing KIF22-GFP T158D in the presence (36%) or absence (32%) of endogenous KIF22 (*Figure 9I*). Expression of KIF22-GFP (10%) or KIF22-GFP T158A

(11%) in the presence of endogenous KIF22 also resulted in a small increase in the percentage of cells with abnormally shaped nuclei compared to control cells expressing GFP (5%) (*Figure 9I*). The expression of KIF22-GFP T158D phenocopies the expression of KIF22-GFP T463D or KIF22-GFP with pathogenic mutations, suggesting that dephosphorylation of T158 contributes to KIF22 inactivation in anaphase.

One model that could explain the observation that structural changes in both the tail and the motor domain of KIF22 disrupt inactivation and cause anaphase chromosome recongression is that these domains physically interact to inactivate the motor, as has been described for other members of the kinesin superfamily (*Blasius et al., 2021*; *Coy et al., 1999*; *Espeut et al., 2008*; *Friedman and Vale, 1999*; *Hammond et al., 2010*; *Hammond et al., 2009*; *Imanishi et al., 2006*; *Ren et al., 2018*; *Verhey and Hammond, 2009*; *Verhey et al., 1998*). To test this model, fluorescently tagged motor domain (1–383) and tail (442–506 or 420–520) fragments were co-expressed in HeLa-Kyoto cells (*Figure 9—figure supplement 3A*). Tail fragments tested excluded both NLSs (*Tahara et al., 2008*) and were cytoplasmic in interphase and mitosis (*Figure 9—figure supplement 3B-C*). Despite containing the second microtubule-binding domain, neither mCh-Tail fragment detectably localized to microtubules (*Figure 9—figure supplement 3B, C*). Immunoprecipitation (IP) was performed to test for interaction between Motor Domain-GFP and mCh-Tail 442–506 or mCh-Tail 420–520. Motor Domain-GFP (molecular weight ~70 kDa) was detected in samples after anti-GFP IP, but neither mCh-Tail 442–506 (~35 kDa) nor mCh-Tail 420–520 (~39 kDa) co-immunoprecipitated with Motor Domain-GFP (*Figure 9—figure supplement 3D*).

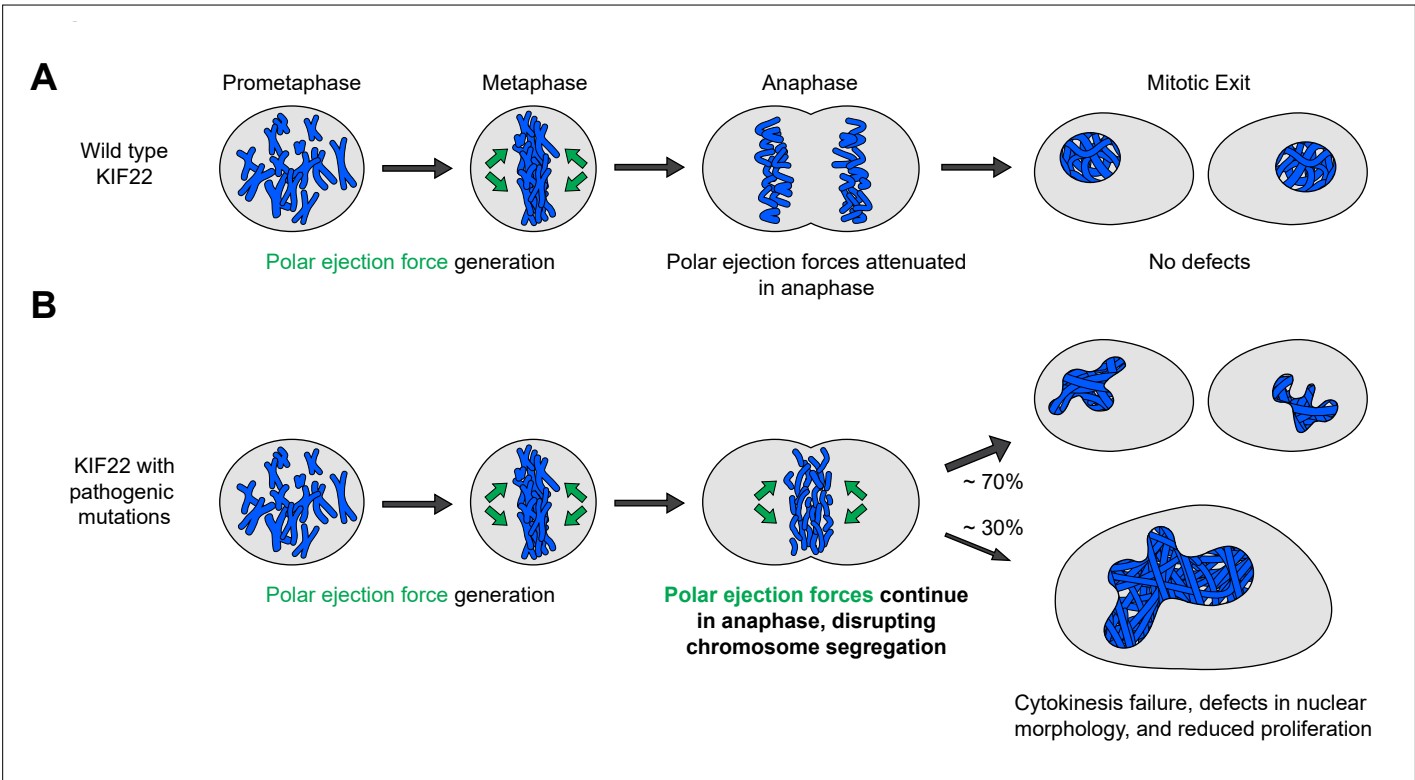

**Figure 10.** Pathogenic mutations disrupt the anaphase, but not prometaphase, function of KIF22. (**A**) Wild-type KIF22 generates polar ejection forces to contribute to chromosome congression and alignment in prometaphase. In anaphase, KIF22 inactivation results in the attenuation of polar ejection forces (green arrows), allowing chromosomes to segregate toward the poles. Daughter cells form regularly shaped nuclei and continue to proliferate. (**B**) In cells expressing KIF22 with pathogenic (P148L, P148S, R149L, R149Q, and V475G) or phosphomimetic (T158D and T463D) mutations, prometaphase proceeds as in cells expressing wild-type motor. Mutant KIF22 is capable of polar ejection force generation. In anaphase, KIF22 fails to inactivate, resulting in continued generation of polar ejection forces, which disrupts anaphase chromosome segregation. Daughter cells exhibit nuclear morphology defects. In about 30% of cells expressing KIF22-GFP R149Q or KIF22-GFP V475G, cytokinesis fails, and proliferation rates are reduced.

## Discussion

We have determined that pathogenic mutations in KIF22 disrupt anaphase chromosome segregation, causing chromosome recongression, nuclear morphology defects, reduced proliferation, and, in a subset of cells, cytokinesis failure. Wild-type KIF22 is inactivated in anaphase (*Soeda et al., 2016*), resulting in an attenuation of polar ejection forces, which allows chromosomes to move toward the spindle poles (*Figure 10A*). The phenotypes we observe in cells expressing KIF22-GFP with pathogenic mutations are consistent with KIF22 remaining active in anaphase (*Figure 10B*). Polar ejection forces could cause recongression by continuing to push chromosomes away from the spindle poles during anaphase A and disrupting spindle elongation during anaphase B. These forces result in aberrant positioning of chromosomes during telophase and cytokinesis, which could cause the nuclear morphology defects and cytokinesis failure we observe in cells expressing mutant KIF22-GFP. Consistent with this model, mimicking phosphorylation of T463 to prevent KIF22 inactivation in anaphase phenocopies the effects of pathogenic mutations. Thus, we conclude that pathogenic mutations result in a gain of KIF22 function, which aligns with findings that KIF22 mutations are dominant in heterozygous patients (*Boyden et al., 2011*; *Min et al., 2011*; *Tüysüz et al., 2015*). The effects of pathogenic mutations on chromosome movements in anaphase are consistent with observations of chromosome recongression in cells with altered CDK1 activity (*Su et al., 2016*; *Wolf et al., 2006*) or altered tail structure (*Soeda et al., 2016*). Our work additionally demonstrates the involvement of the motor domain α2 helix in this process and the consequences of recongression on cytokinesis, daughter cell nuclear morphology, and proliferation.

Mutations in both the motor domain (P148L, P148S, R149L, and R149Q) and the coiled-coil domain (V475G) of KIF22 disrupt chromosome segregation in a manner consistent with a failure of KIF22 inactivation in anaphase. Additionally, mimicking phosphorylation of T158 in the motor domain or T463 in the tail also disrupts chromosome segregation. These findings demonstrate that the motor domain α2 helix participates in the process of KIF22 inactivation, adding to studies that demonstrate that deletion of the tail microtubule-binding domain and deletion or disruption of the coiled-coil domain prevent the inactivation of KIF22 in anaphase (*Soeda et al., 2016*). Phosphorylation of KIF22 T158 has been detected in mitotic cells, but the relative phosphorylation levels of this residue at different times in mitosis are not known (*Olsen et al., 2010*). Further studies are needed to determine the time course of T158 phosphorylation, whether this phosphorylation regulates KIF22 function in mitosis, and if T158 and T463 phosphorylation control KIF22 activity independently or via the same mechanism.

The physical mechanism of KIF22 inactivation is unknown, and our results can be interpreted in the context of several models, which are not mutually exclusive. Previous work has proposed that the tail of KIF22 may interact with microtubules to suspend polar ejection force generation, as KIF22 inactivation requires the tail second microtubule-binding and coiled-coil domains (*Soeda et al., 2016*). Mimicking phosphorylation of T463 disrupts inactivation (*Soeda et al., 2016*). Dephosphorylation of T463 could facilitate tail-microtubule interaction, or charge change at T463 could disrupt the structure of the microtubule-binding or coiled-coil domains. The interaction between the tail of KIF22 and the microtubule may not be strong or long-lasting, as deletion of the tail microtubule-binding domain does not alter the velocity of KIF22 (*Shiroguchi et al., 2003*), and KIF22 tail fragments containing the second microtubule-binding domain do not localize to spindle microtubules. In the framework of a tail-microtubule interaction inactivating KIF22, the mutation in the tail (V745G) could disrupt anaphase chromosome segregation by altering this interaction with microtubules. Whether or how the α2 helix could contribute to this mechanism is less clear. The α2 helix faces away from the surface of the microtubule, and we would not predict that mutations in this structure would directly alter the association of the motor domain with the microtubule. It is possible that this region of the motor domain could facilitate or strengthen an interaction between the tail and the microtubule surface indirectly.

Alternatively, given that mutations in the tail and motor domain of KIF22 both disrupt chromosome segregation, the tail and motor domain may interact to inactivate the motor. Head-tail autoinhibition is a known regulatory mechanism of other members of the kinesin superfamily (*Blasius et al., 2021*; *Coy et al., 1999*; *Espeut et al., 2008*; *Friedman and Vale, 1999*; *Hammond et al., 2010*; *Hammond et al., 2009*; *Imanishi et al., 2006*; *Ren et al., 2018*; *Verhey and Hammond, 2009*; *Verhey et al., 1998*), and disruption of autoinhibition can be a mechanism of disease pathogenesis (*Asselin et al., 2020*; *Bianchi et al., 2016*; *Blasius et al., 2021*; *Cheng et al., 2014*; *Pant et al., 2022*; *van der Vaart et al., 2013*). Mutations in either the tail or motor domain could disrupt this interaction,

preventing KIF22 inactivation in anaphase. Dephosphorylation of both T463 in the tail and T158 in the motor domain could facilitate this interaction. Co-IP experiments did not demonstrate an interaction between the motor domain and tail of KIF22 under the conditions of our assays. However, it is possible that a transient head-tail interaction may not be detectable by IP. Thus, further studies are needed to rule out an inactivating interaction between these domains.

Rather than physically interacting with the motor domain, it is also possible that structural changes in the tail of KIF22 could have allosteric effects on the motor domain. An allosteric mechanism by which conformational changes are propagated down the stalk to the motor domain has recently been proposed to contribute to the inactivation of kinesin-1 motors by kinesin light chain, which binds the tail (*Chiba et al., 2021*). KIF22 inactivation may be caused by altered motor domain mechanochemistry, which changes in the tail could affect allosterically and modification of α2 could affect directly. This could explain the effect of tail and motor domain mutations, as well as the effects of mimicking tail and motor domain phosphorylation, on KIF22 activity.

An additional consideration is that pathogenic mutations may affect the inactivation of KIF22 in anaphase by altering phosphoregulation of KIF22 activity. If mutations prevented the dephosphorylation of T158 and T463 in anaphase this could cause anaphase recongression. However, addition of a phosphonull T463A mutation to KIF22 with coiled-coil or microtubule-binding domain deletions does not rescue anaphase chromosome recongression defects (*Soeda et al., 2016*), suggesting that the role of the KIF22 tail in motor inactivation is not only to facilitate dephosphorylation of T463. Future studies using structural approaches will be required to distinguish between possible mechanisms of KIF22 inactivation.

The regulation of the motor domain α2 helix in KIF22 inactivation may inform our understanding of additional kinesin motors, as amino acids P148 and R149 are conserved in a number of members of the kinesin superfamily (*Figure 1D*). Similarly, phosphorylation or acetylation of amino acids in the α2 helix has been reported for members of the kinesin-3 (KIF13A S134) (*Dephoure et al., 2008*), kinesin-5 (KIF11 Y125, K146) (*Bickel et al., 2017*; *Choudhary et al., 2009*), kinesin-6 (KIF20B S182 and KIF23 S125) (*Hegemann et al., 2011*; *Sharma et al., 2014*; *Shiromizu et al., 2013*), and kinesin-14 (KIFC3 S557) (*Sharma et al., 2014*) families. Phosphorylation of Y125 (*Bickel et al., 2017*) and acetylation of K146 (*Muretta et al., 2018*) in KIF11 (Eg5) have been shown to modulate motor activity, and the functions of the remaining reported post-translation modifications in the α2 helix are yet to be characterized. Acetylation of KIF11 at K146 increases the stall force of the motor and slows anaphase spindle pole separation (*Muretta et al., 2018*). This post-translational modification represents a mechanism by which the activity of KIF11 could be regulated at the metaphase to anaphase transition to generate sliding forces for spindle assembly in prometaphase and control spindle pole separation in anaphase, analogous to how post-translational modifications of KIF22 regulate motor activity to ensure both chromosome congression and alignment in prometaphase and chromosome segregation in anaphase.

While chromosomes in some cells, particularly those expressing KIF22-GFP at high levels, completely failed to segregate and decondensed in the center of the spindle, most cells demonstrated chromosome recongression wherein poleward motion of chromosomes begins, but then chromosomes switch direction and move anti-poleward. These dynamics may be due to differences in microtubule density closer to the poles compared to the center of the spindle. This model is consistent with work demonstrating that in monopolar spindles, poleward movement of chromosomes is limited by chromosomes reaching a threshold density of microtubules at which polar ejection forces are sufficient to cause chromosomes to switch to anti-poleward movement (*Cassimeris et al., 1994*). We observed that chromosomes on the periphery of the spindle remain closer to the poles while central chromosomes are pushed further away from the poles during recongression in cells expressing KIF22-GFP with pathogenic mutations. This could also be explained by the central chromosomes encountering a higher density of microtubules, and KIF22 bound to these chromosomes therefore generating higher levels of polar ejection forces. In addition, this mechanism is consistent with observations that oscillations of peripheral chromosomes are reduced compared to chromosomes at the center of the spindle (*Cameron et al., 2006*; *Cimini et al., 2004*; *Civelekoglu-Scholey et al., 2013*; *Stumpff et al., 2008*), which could also be explained by reduced peripheral microtubule density limiting peripheral polar ejection force generation.

Our assessment of the relative trajectories of chromosomes, centromeres, and spindle poles offers insight into the relative magnitudes of polar ejection forces and other anaphase forces. Expression of

KIF22-GFP with pathogenic mutations did not alter the distance between centromeres and spindle poles, indicating that while anaphase polar ejection forces altered the position of chromosome arms within the spindle, these forces were not sufficient to prevent the shortening of k-fibers. However, the expression of mutant KIF22-GFP did alter the movements of the spindle poles, allowing assessment of the relative magnitude of polar ejection forces compared to the forces generated by the sliding of antiparallel spindle microtubules to separate the spindle poles in anaphase (*Brust-Mascher et al., 2004*; *Fu et al., 2009*; *Nislow et al., 1992*; *Sawin et al., 1992*; *Straight et al., 1998*; *Tanenbaum et al., 2009*; *van Heesbeen et al., 2014*; *Vukušić et al., 2019*; *Vukušić et al., 2021*). In cells expressing mutant KIF22-GFP, spindle pole separation stalled, and poles moved closer to one another during anaphase chromosome recongression. This suggests that the polar ejection forces collectively generated by mutant KIF22 motors are of greater magnitude than the forces sliding the spindle poles apart during anaphase B. Although it is important to note that this phenotype was observed with moderate overexpression of mutant KIF22, the observed effects on spindle pole separation underscore the importance of KIF22 inactivation, and imply that reducing polar ejection forces is required for both anaphase A and anaphase B. This force balance may differ between cell types, as tail domain deletions that alter chromosome movements do not disrupt anaphase B in mouse oocyte meiosis (*Soeda et al., 2016*).

Patients with mutations in KIF22 exhibit defects in skeletal development. The pathology observed in the patient heterozygous for the V475G mutation differs from those seen in SEMDJL2 patients with motor domain mutations (*Figure 1E and F*; *Boyden et al., 2011*; *Min et al., 2011*; *Tüysüz et al., 2015*). However, a meaningful comparison of pathologies between patients is limited both by the fact that only a single patient with a mutation in the tail of KIF22 has been identified, and by the considerable variation in clinical presentation between patients with motor domain mutations, even between patients with the same point mutation (*Boyden et al., 2011*; *Min et al., 2011*; *Tüysüz et al., 2015*). The defects in chromosome segregation we observed in cells expressing mutant KIF22-GFP may contribute to skeletal developmental pathogenesis. Mutations could cause reduced proliferation of growth plate chondrocytes, which in turn could limit bone growth. Disrupting cytokinesis in the growth plate causes shorter bones and stature in mice (*Gan et al., 2019*), and mutations in KIF22 could affect development via this mechanism. The presence of pathologies in other cartilaginous tissues, including the larynx and trachea, in patients with mutations in the motor domain of KIF22 (*Boyden et al., 2011*) is also consistent with a disease etiology based in aberrant chondrocyte proliferation. Defects in mitosis could result in tissue-specific patient pathology based on differences in force balance within anaphase spindles in different cell types arising from different expression or activity levels of mitotic force generators or regulators. Growth plate chondrocytes, particularly, are organized into columns and must divide under geometric constraints (*Dodds, 1930*), which could increase sensitivity to anaphase force imbalances. Additionally, we cannot exclude the possibility that these mutations may affect the function of interphase cells, which could affect development via a mechanism independent from the effects of the mutations on mitosis. Future work will be required to distinguish among these possible explanations.

## Materials and methods

### Patient assessment

Clinical exome sequencing was performed by the Department of Laboratory Medicine and Pathology at Mayo Clinic in Rochester, Minnesota, USA as previously described (*Cousin et al., 2019*). Carbohydrate deficient transferrin testing for congenital disorders of glycosylation was performed at Mayo Clinic Laboratories, Rochester, Minnesota, USA (*Lefeber et al., 2011*).

### Cell culture

Human HeLa-Kyoto (RRID:CVCL_1922, gift of Ryoma Ohi, University of Michigan) and RPE-1 cell lines (ATCC #CRL-4000, RRID:CVCL_4388) were grown in Minimum Essential Media α (Gibco #12561-056) supplemented with 10% fetal bovine serum (FBS; Gibco #16000-044) at 37°C with 5% $CO_2$. Cell lines were validated by short tandem repeat (STR) DNA typing using the Promega GenePrint 10 System according to the manufacturer's instructions (Promega #B9510). Cells were cryopreserved in Recovery Cell Culture Freezing Medium (Gibco #12648-010). HeLa-Kyoto and RPE-1 acceptor cell lines for

recombination (both gifts from Ryoma Ohi, University of Michigan) were maintained in media supplemented with 10 µg/ml blasticidin (Thermo Fisher Scientific #R21001).

Transfection siRNA transfection was performed using Lipofectamine RNAiMax Transfection Reagent (Thermo Fisher Scientific #13778150) in Opti-MEM Reduced Serum Media (Gibco #31985-062). KIF22 was targeted for siRNA-mediated depletion using a Silencer Validated siRNA (Ambion #AM51331, sense sequence GCUGCUCUCUAGAGAUUGCTT). Control cells were transfected with Silencer Negative Control siRNA #2 (Ambion #AM4613). DNA transfections were performed using Lipofectamine LTX (Thermo Fisher Scientific #15338100) in Opti-MEM Reduced Serum Media (Gibco #31985-062).

## Plasmids

Plasmids related to the generation of inducible cell lines are described in *Table 2*. A C-terminally tagged KIF22-GFP plasmid was constructed by adding EcoRI and KpnI sites to the KIF22 open reading frame (from pJS2161; *Stumpff et al., 2012*), performing a restriction digest, and ligating the products into a digested pEGFP-N2 vector (Clontech) (pAT4206). Site-directed mutagenesis was performed to add silent mutations for siRNA resistance (pAT4226). The open reading frame from pAT4226 and the pEM791 vector (*Khandelia et al., 2011*) were amplified and combined using Gibson Assembly (New England BioLabs) to generate a plasmid for recombination-mediated construction of inducible cell lines (pAT4250). Site-directed mutagenesis was performed on pAT4250 to generate plasmids encoding KIF22-GFP P148L, P148S, R149L, R149Q, V475G, T463D, T463A, T134D, T158D, and T158A for recombination. A plasmid encoding KIF22-GFP T134A for recombination was generated using Gibson Assembly of a synthesized DNA fragment (Thermo Fisher Scientific) and pAT4250. A plasmid encoding Motor Domain-GFP for recombination was generated from pAT4250 by deletion. Plasmids encoding mCh-Tail 442–506 and mCh-Tail 420–520 were generated using Gibson Assembly of pAT4250 and mCh-CAAX. See *Table 2* for primer sequences. Plasmids generated from this study are described in *Table 2* and available on request from the authors.

The mCh-CAAX plasmid was a gift from Alan Howe (University of Vermont). The mCh-NLS plasmid was generated by Michael Davidson and obtained from Addgene (mCh-Nucleus-7, #55110). The pericentrin-RFP plasmid (*Gillingham and Munro, 2000*) was a gift from Sean Munro (MRC Laboratory of Molecular Biology). The CENPB-mCh plasmid (*Liu et al., 2010*) was generated by Michael Lampson and obtained from Addgene (#45219).

## Generation of inducible cell lines

Inducible cell lines were generated using recombination-mediated cassette exchange as previously described (*Khandelia et al., 2011*). Briefly, plasmids (see *Table 2*) encoding siRNA-resistant KIF22-GFP constructs were cotransfected with a plasmid encoding nuclear-localized Cre recombinase (pEM784) into HeLa-Kyoto (*Sturgill et al., 2016*) or RPE-1 acceptor cells using Lipofectamine LTX transfection (Thermo Fisher Scientific #15338100). For HeLa-Kyoto cell lines, 24 hr after transfection cells were treated with 1 µg/mL puromycin (Thermo Fisher Scientific #A11139-03) for 48 hr, then 2 µg/ml puromycin for 48 hr for more stringent selection, and finally 1 µg/ml puromycin until puromycin-sensitive cells were eliminated. Selection of RPE-1 cells was accomplished via treatment with 5 µg/ml puromycin for 48 hr beginning 24 hr after transfection, then 10 µg/ml puromycin for 48 hr, and finally 5 µg/ml puromycin until puromycin-sensitive cells were eliminated. Inducible cell lines were maintained in puromycin (HeLa-Kyoto 1 µg/ml, RPE-1 5 µg/ml) for continued selection. To confirm the sequence of inserted DNA in the selected cell populations, genomic DNA was extracted using the QIAmp DNA Blood Mini Kit (Qiagen #51106) and subjected to sequencing (Eurofins). Expression of inserted DNA sequences was induced via treatment with 2 µg/ml doxycycline (Thermo Fisher Scientific #BP26531). Cell lines generated from this study are available on request from the authors.

## Immunoprecipitation and western blotting

HeLa-Kyoto cells were lysed in 10 mM tris buffer, pH 7.5, with 150 mM NaCl, 0.5 mM EDTA, 0.5% Triton X-100 (Sigma-Aldrich #93443), and 1× Halt Protease and Phosphatase Inhibitor (Thermo Fisher Scientific #78442) on ice. Anti-GFP IP was performed using GFP-Trap nanobody-coated magnetic particles (ChromoTek #GTD-20). Samples were separated by electrophoresis on 4–15% or 4–20% trisglycine polyacrylamide gels (Bio-Rad #4561083 or #4561093) and transferred to polyvinylidene fluoride membranes (Bio-Rad #162-0261). Membranes were blocked in Intercept TBS Blocking Buffer (LI-COR

**Table 2.** Plasmids used in this study.

| Plasmid | Description | Primers (5' to 3', Fw: Forward, Rev: Reverse) | Source |
|---|---|---|---|
| pEM784 | nlCre recombinase | NA | Khandelia 2011 PMID 21768390 |
| pEM791 | EGFP for recombination | NA | Khandelia 2011 PMID 21768390 |
| pJS2161 | GFP-KIF22 | NA | Stumpff 2012 PMID 22595673 |
| pAT4206 | KIF22-GFP | Fw: TACGTGGAATTCCACCATGGCCGCGGGGCGGCTCGA Rev: GTGACTGGTACCTGGAGGCGCCACAGCGCTGGC | This study |
| pAT4226 | KIF22-GFP, siRNA resistant | Fw:pGGGCATGGACAGCTGCTCACTCGAAATCGCTAACTGGAGGAACCAC Rev:pGTGGTTCCTCCAGTTAGCGATTTCGAGTGAGCAGCTGTCCATGCCC | This study |
| pAT4250 | KIF22-GFP, siRNA resistant, for recombination | Fragment Fw: CTGGGCACCACCATGGCCGCG Fragment Rev: GCTAGCTCGATTACTTGTACAGCTCGTCCATGCC Vector Fw: GTACAAGTAATCGAGCTAGCATATGGATCCATATAACT Vector Rev: CATGGTGGTGCCCAGTGCCTCACGACC | This study |
| pAT4251 | KIF22-GFP R149Q | Fw: GGGGTGATCCCGCAGGCTCTCATGGAC Rev: GTCCATGAGAGCCTGCGGGATCACCCC | This study |
| pAT4258 | KIF22-GFP V475G | Fw: TGCTAATGAAGACAGGAGAAGGAGAAGGACCT Rev: AGGTCCTTCTCCTTCTCCTGTCTTCATTAGCA | This study |
| pAT4260 | KIF22-GFP T463D | Fw: CCCCTCTGTTGAGTGACCCAAAGCGAGAGC Rev: GCTCTCGCTTTGGGTCACTCAACAGAGGGG | This study |
| pAT4261 | KIF22-GFP T463A | Fw: CCTCTGTTGAGTGCCCCAAAGCGAG Rev: CTCGCTTTGGGGCACTCAACAGAGG | This study |
| pAT4264 | KIF22-GFP R149L | Fw: GGGGTGATCCCGCTGGCTCTCATGGAC Rev: GTCCATGAGAGCCAGCGGGATCACCC | This study |
| pAT4269 | KIF22-GFP P148L | Fw: CCTGGGGGTGATCCTCGGGGCTCTCATG Rev: CATGAGAGCCCGCAGGATCACCCCAGG | This study |
| pAT4270 | KIF22-GFP P148S | Fw: CTGGGGTGATCTCGGGGCTCTCATG Rev: CATGAGAGCCCGCGAGATCACCCCAG | This study |
| pSS4279 | KIF22-GFP T134A | Fragment Fw: AGCTGCTCACTCGAAATCGC Fragment Rev: AGTCTTTCTCGGATTACCAGG Vector Fw: CCTGGTAATCCGAGAAGACT Vector Rev: GCGATTTCGAGTGAGCAGCT | This study |
| pSS4281 | KIF22-GFP T134D | Fw: CAGGAGCTGGGAAGGATCACACAAATGCTGGGC Rev: GCCCAGCATTGTGTGATCCTTCCCAGCTCCTG | This study |

*Table 2 continued on next page*

Table 2 continued

| Plasmid | Description | Primers (5' to 3', Fw: Forward, Rev: Reverse) | Source |
|---|---|---|---|
| pNA4285 | KIF22-GFP T158A | Fw: AGCTCGCAAGGGAGGAGGGGTG Rev: GAGTACCTGGAGGACGTCGA | This study |
| pNA4284 | KIF22-GFP T158D | Fw: CCTCCTGCAGCTCAGGGAGGAGGGGTG Rev: CACCCTCCTCCCTGAGCTGCAGGAGG | This study |
| pAT4291 | mCh-Tail 442–506 | Fragment Fw: ccggactcagatctcgaggacgcctcctcagcttggaccg Fragment Rev: ctgattatgatcagttatcgttcctttcctcagccttctg Vector Fw: aggctgaggaaaagagaacagataactgatcataatcagccatac Vector Rev: cggtccaagctgaggaggcgtcctcgagatctgagtccgg | This study |
| pAT4292 | mCh-Tail 420–520 | Fragment Fw: ccggactcagatctcgaggagcctgcctcccagaaact Fragment Rev: ctgattatgatcagttatctgactgtgcgatgtgaaaggg Vector Fw: ccctttcacatcgcacagtcagataactgatcataatcagccatac Vector Rev: agtttctgggaggcagaggtcctcgagatctgagtccgg | This study |
| pAT4294 | Motor Domain-GFP 1–383 | Fw: aggtaccgcgggcccgggat Rev: ccaatgagagagcctgcagcctcatgccttg | This study |

#927-60001) and blotted using rabbit anti-GFP (1:1000, Invitrogen #A11122, RRID:AB_221569) or rabbit anti-mCherry (1:1000, Abcam #167453, RRID:AB_2571870) primary antibodies incubated overnight at 4°C. Blots were incubated with goat anti-Rabbit IgG DyLight 800 secondary antibody (1:10,000, Thermo Fisher Scientific #SA5-10036, RRID:AB_2556616) for 1 hr at room temperature. Imaging was performed using an Odyssey CLX system (LI-COR) and images were processed using Image Studio Lite (LI-COR, version 5.2.5).

## Immunofluorescence

For fixed cell imaging, cells were grown on 12 mm glass coverslips in 24-well plates. Cells were fixed in 1% paraformaldehyde in ice-cold methanol for 10 min on ice. Cells were blocked for 1 hr using 20% goat serum (Gibco #16210-064) in antibody dilution buffer (AbDil, 1% bovine serum albumin (Sigma-Aldrich #B4287), 0.1% Triton X-100 (Sigma-Aldrich #93443), 0.02% sodium azide (Thermo Fisher Scientific #BP9221) in TBS) and incubated with the following primary antibodies for 1 hr at room temperature: mouse anti-α-tubulin (DM1α) 1:500 (MilliporeSigma #T6199, RRID:AB_477583), rat anti-tubulin clone YL1/2 1:1,500 (MilliporeSigma #MAB1864, RRID:AB_2210391), rabbit anti-KIF22 1:500 (GeneTex #GTX112357, RRID:AB_11166142), mouse anti-centrin 1:500 (MilliporeSigma #04-1624, RRID:AB_10563501), or rabbit anti-GFP 1:1000 (Invitrogen #A11121, RRID:AB_221567). Cells were incubated with secondary antibodies conjugated to AlexaFluor 488, 594, or 647 (Invitrogen Molecular Probes #A11034 RRID:AB_2576217, A11037 RRID:AB_2534095, A21245 RRID:AB141775, A11029 RRID:AB_2534088, A11032 RRID:AB_2534091, A21236 RRID:AB_2535805, and A11007 RRID:AB_141374) for 1 hr at room temperature. All incubations were performed on an orbital shaker. Coverslips were mounted on slides using Prolong Gold mounting medium with DAPI (Invitrogen Molecular Probes #P36935).

## Microscopy

Images were acquired using a Nikon Ti-E or Ti-2E inverted microscope driven by NIS Elements software (Nikon Instruments). Images were captured using a Clara cooled charge-coupled device camera (Andor) or Prime BSI scientific complementary metal-oxide-semiconductor camera (Teledyne Photometrics) with a Spectra-X light engine (Lumencore). Samples were imaged using Nikon objectives Plan Apo 40× 0.95 numerical aperture (NA), Plan Apo $\lambda$ 60× 1.42 NA, and APO 100× 1.49 NA. For live imaging, cells were imaged in $CO_2$-independent media (Gibco #18045-088) supplemented with 10% FBS (Gibco #16000-044) in a 37°C environmental chamber. Images were processed and analyzed using ImageJ/FIJI (*Schindelin et al., 2012*; *Schneider et al., 2012*).

## KIF22-GFP expression level quantitation

HeLa-Kyoto or RPE-1 cells were treated with 2 µg/ml doxycycline to induce expression and transfected with control or KIF22 siRNA approximately 24 hr prior to fixation. Metaphase cells were imaged for measurement of KIF22 expression levels. Measurements of KIF22 immunofluorescence intensity were made in a background region of interest (ROI) containing no cells and an ROI representing the chromosomes, identified by thresholding DAPI signal. The mean background subtracted KIF22 signal on the chromosomes was calculated by subtracting the product of the mean background intensity and the chromosome ROI area from the chromosome ROI integrated density and dividing by the area of the chromosome ROI. KIF22 intensities were normalized to the mean KIF22 intensity in control cells (control knockdown, uninduced) in each experimental replicate. Since mutations at T134 alter the localization of KIF22, measurements of KIF22 intensity in T134 cell lines and corresponding control cells were made using a circular ROI enclosing the spindle, identified by tubulin signal. The same background subtraction and normalization approaches were then used with these measurements.

## Metaphase chromosome spreads

RPE-1 cells were grown in 60 mm dishes for approximately 24 hr. Media were exchanged to fresh growth media for 2 hr to promote mitosis. Cells were arrested in 0.02 µg/ml colcemid (Gibco KaryoMAX #15212012) for 3 hr at 37°C, then trypsinized, pelleted, and gently re-suspended in 500 µl media. 5 ml 0.56% KCl hypotonic solution was added dropwise to the cell suspension, which was then incubated for 15 min in a 37°C water bath. Cells were pelleted, gently resuspended, and fixed via the addition of 1 ml ice-cold 3:1 methanol:glacial acetic acid. Cells were pelleted and resuspended

in fixative an additional three times, then stored at –20°C. Metaphase chromosome spreads were prepared by humidifying the surface of glass slides by exposing them to the steam above a 50°C water bath, placing the slides at an angle relative to the work surface, and dropping approximately 100 µl of ice-cold cell suspension onto the slide from a height of approximately 1 ft. Slides were dried on a hot plate, then covered with Prolong Gold mounting medium with DAPI (Invitrogen Molecular Probes #P36935) and sealed.

## Fluorescence recovery after photobleaching

HeLa-Kyoto cells were seeded in glass-bottom 35 mm dishes (Greiner Bio-One #627975 and #627965) and treated with 2 µg/ml doxycycline to induce expression 18–24 hr before imaging. Cells were imaged at 5-s intervals for 25 s before bleaching, photobleached using a point-focused 405 nm laser, and imaged at 20-s intervals for 10 min after bleaching. Fluorescence intensities in bleached, unbleached, and background regions of each frame were measured using a circular ROI, area 0.865 µm². For interphase and metaphase cells, unbleached measurements were made on the opposite side of the nucleus or chromosome mass as the bleached measurements. For anaphase cells, one segregating chromosome mass was bleached, and unbleached measurements were made on the opposite chromosome mass. Background intensities, measured in cell-free area, were subtracted from bleached and unbleached intensities. Background-subtracted intensities were normalized to the intensity of the first frame imaged.

## Polar ejection force assay

HeLa-Kyoto cells were treated with 2 µg/ml doxycycline to induce expression and transfected with control or KIF22 siRNA approximately 24 hr prior to fixation. Cells were arrested in 100 µM monastrol (Selleckchem #S8439) for 2–3 hr before fixation. Monopolar mitotic cells oriented perpendicular to the coverslip were imaged at the focal plane of the spindle pole for polar ejection force measurements. A circular ROI with a 12.5 µm radius was centered around the spindle pole of each cell, and the radial profile of DAPI signal intensity at distances from the pole was measured (Radial Profile Plot plugin; https://imagej.nih.gov/ij/plugins/radial-profile.html). The distance from the pole to the maximum DAPI signal was calculated for each cell as a measure of relative polar ejection forces (*Thompson et al., 2022*).

## Analyses of anaphase chromosome segregation

HeLa-Kyoto or RPE-1 cells were treated with 2 µg/ml doxycycline to induce expression approximately 18 hr before imaging. For HeLa-Kyoto cells, media ~ exchanged to $CO_2$-indpendent media containing 2 µg/ml doxycycline and 100 nM SiR-Tubulin (Spirochrome #SC002) approximately 1–1.5 hr before imaging. For RPE-1 cells, media were exchanged to $CO_2$-indpendent media containing 2 µg/ml doxycycline, 20–100 nM SiR-Tubulin (Spirochrome #SC002), and 10 µM verapamil (Spirochrome #SCV01) approximately 1.5–3 hr before imaging. Cells were imaged at 1-min time intervals. Distances between segregating chromosome masses were measured by plotting the KIF22-GFP signal intensity along a line drawn through both spindle poles (macro available at https://github.com/StumpffLab/Image-Analysis; *Stumpff, 2021*). This data set was split at the center distance to generate two plots, each representing one half-spindle/segregating chromosome mass. The distance between the maximum of each intensity plot was calculated using MATLAB (MathWorks, version R2018a) (script available at https://github.com/StumpffLab/Image-Analysis). To assess the broadness of segregating chromosome masses in cells expressing KIF22-GFP T463A, a Gaussian curve was fit to the same intensity plots and the full width at half maximum was calculated in MATLAB.

To measure the movements of spindle poles and kinetochores in anaphase, HeLa-Kyoto cells were seeded in glass-bottom 24-well plates (Cellvis #P24-1.5H-N) and cotransfected with PCM-RFP and mCh-CENPB using Lipofectamine LTX (Thermo Fisher Scientific #15338100) approximately 24 hr before imaging. Cells were treated with 2 µg/ml doxycycline to induce expression approximately 12–18 hr before imaging. Cells were imaged at 20-s time intervals. To more clearly visualize spindle poles and kinetochores, images of PCM-RFP and mCh-CENPB signal were background subtracted by duplicating each frame, applying a Gaussian blur (Sigma-Aldrich 30 pixels), and subtracting this blurred image from the original. For each frame, a line was drawn between spindle poles (PCM-RFP signal) to measure the distance between them, and the intensity of KIF22-GFP and mCh-CENPB along

this line was plotted. These data sets were split at the center distance to generate two plots, and the distance between plot maxima and the distance from maxima to the spindle poles were calculated using MATLAB (scripts available at https://github.com/StumpffLab/Image-Analysis).

### Assessment of cytokinesis failure

To visualize cell boundaries, HeLa-Kyoto cells were transfected with mCh-CAAX using Lipofectamine LTX approximately 24–32 hr before imaging and treated with 2 µg/ml doxycycline approximately 8 hr before imaging. Cells were imaged at 3-min intervals. Cells were scored as failing cytokinesis if the product of mitosis was a single cell with a single boundary of mCh-CAAX signal.

### Nuclear morphology quantification

HeLa-Kyoto or RPE-1 cells were treated with 2 µg/ml doxycycline to induce expression approximately 24 hr before fixation. Nuclear solidity was measured for each interphase nucleus in each imaged field. The fifth percentile of solidity for control cells (transfected with control siRNA and expressing GFP) was used as a threshold below which nuclear solidity was considered abnormal.

To assess the ability of nuclei to retain nuclear-localized proteins, cells were transfected with mCh-NLS using Lipofectamine LTX approximately 24–32 hr before imaging and treated with 2 µg/ml doxycycline approximately 8 hr before imaging. Cells were imaged at 3-min intervals during and after division, and the presence of mCh-NLS signal in all nuclear structures (KIF22-GFP positive regions) was assessed.

### Assessment of spindle dependence of nuclear morphology defects

To assess whether nuclear morphology defects caused by KIF22 depend on force generation within the mitotic spindle, cells were treated with 2 µg/ml doxycycline approximately 8 hr before imaging, SPY595-DNA (1× per manufacturer's instructions) (Spirochrome #SC301) approximately 1.5–2 hr before imaging, and 500 nM nocodazole (Selleckchem #S2775) and 900 nM reversine (Cayman Chemical #10004412) approximately 0.5–1 hr before imaging. Cells were imaged at 5-min intervals. Nuclear solidity was measured 15 min before chromosome condensation and 100 min after chromosome decondensation.

### Proliferation assay

HeLa-Kyoto cells were seeded in a 96-well plate and treated with 2 µg/ml doxycycline to induce expression or transfected with KIF22 siRNA approximately 8 hr before the first assay time point. Automated bright field imaging using a Cytation 5 Cell Imaging Multi-Mode Reader (BioTek) (4× Plan Fluorite 0.13 NA objective; Olympus) driven by Gen5 software (BioTek) was used to measure cell proliferation (*Marquis et al., 2021*). Images were collected every 4 hr for 96 hr. Gen5 software was used to process images and count the number of cells in each imaged field. Cell counts were normalized to the cell count in the first image acquired at time 0. Only wells with first frame cell counts between 10,000 and 20,000 were analyzed to account for the effects of cell density. Fold change at 96 hr was calculated by dividing the cell count at 96 hr by the cell count at time 0. Predicted cell counts at 48 hr were calculated using an experimentally determined doubling time of 20.72 hr for the control case where all cells divide ($Cells_T = 2^{\left(\frac{T}{20.72}\right)}$), the case where nuclear morphology defects limit proliferation and 60% of cells do not divide ($Cells_T = 1.4^{\left(\frac{T}{20.72}\right)}$), and the case where cytokinesis *Table 1* failure limits proliferation and 30% of cells do not divide ($Cells_T = 1.7^{\left(\frac{T}{20.72}\right)}$).

### Statistical analyses

Statistical tests were performed using GraphPad Prism software (GraphPad Software, Inc), version 9.2.0. Specific statistical tests and n values for reported data are indicated in the figure legends. All data represent a minimum of three independent experiments.

## Acknowledgements

This work was supported by NIH F31AR074887 to AFT, NIH R01GM130556 to JKS, NIH R01GM121491 to JKS, NIH R35GM144133 to JKS, and the Ballenger Ventures Fund for Research Excellence. We thank the Mayo Clinic Center for Individualized Medicine (CIM) for supporting this research through the CIM

Investigative and Functional Genomics program. We thank Alan Howe for the mCh-CAAX plasmid and Ryoma Ohi for reagents and acceptor cells for recombination-mediated cassette exchange. We thank Rachel Stadler for technical assistance with data analysis and thank Laura Reinholdt and Matthew Warman for constructive discussions regarding this work.

## Additional information

### Funding

| Funder | Grant reference number | Author |
|---|---|---|
| National Institutes of Health | F31AR074887 | Alex F Thompson |
| National Institutes of Health | R01GM130556 | Jason Stumpff |
| National Institutes of Health | R01GM121491 | Jason Stumpff |
| National Institutes of Health | R35GM144133 | Jason Stumpff |

The funders had no role in study design, data collection and interpretation, or the decision to submit the work for publication.

### Author contributions

Alex F Thompson, Conceptualization, Data curation, Formal analysis, Funding acquisition, Investigation, Methodology, Writing - original draft, Writing – review and editing; Patrick R Blackburn, Data curation, Formal analysis, Investigation, Writing – review and editing; Noah S Arons, Sarah N Stevens, Formal analysis, Investigation; Dusica Babovic-Vuksanovic, Formal analysis, Investigation, Writing – review and editing; Jane B Lian, Conceptualization, Supervision, Writing – review and editing; Eric W Klee, Conceptualization, Formal analysis, Writing – review and editing; Jason Stumpff, Conceptualization, Funding acquisition, Methodology, Writing - original draft, Writing – review and editing

### Author ORCIDs

Alex F Thompson ![ORCID] http://orcid.org/0000-0003-4316-7532
Patrick R Blackburn ![ORCID] http://orcid.org/0000-0003-0658-1275
Eric W Klee ![ORCID] http://orcid.org/0000-0003-2946-5795
Jason Stumpff ![ORCID] http://orcid.org/0000-0003-0392-1254

### Decision letter and Author response

Decision letter https://doi.org/10.7554/eLife.78653.sa1
Author response https://doi.org/10.7554/eLife.78653.sa2

## Additional files

### Supplementary files

- Transparent reporting form
- MDAR checklist

### Data availability

All data generated or analyzed during this study are included in the manuscript and supporting files. Source Data files have been provided for Figure 2, Figure 2- Figure Supplement 1, Figure 3, Figure 4, Figure 4- Figure Supplement 1, Figure 5, Figure 6, Figure 6- Figure Supplement 1, Figure 7, Figure 8, Figure 8- Figure Supplement 1, Figure 9, Figure 9- Figure Supplement 1, Figure 9- Figure Supplement 2, and Figure 9- Figure Supplement 3.

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
