## [Editor Report]

This article analyzes the mechanism of human pathogenicity linked to point mutations in the chromokinesin Kid/Kif22, that cause abnormal skeletal development. The authors show the mutations do not cause a loss of function. Instead, they are dominant negative and fail to inactivate the Kif22 motor, resulting in appropriate force generation of the motor during anaphase. This work highlights that the loss of regulation of kinesin motors in mitosis can disrupt cell division at the cellular scale and human pathogenesis.

---

## [Decision Letter]

**Decision letter after peer review:**

[Editors’ note: the authors submitted for reconsideration following the decision after peer review. What follows is the decision letter after the first round of review.]

Thank you for submitting the paper "Pathogenic mutations in the chromokinesin KIF22 disrupt anaphase chromosome segregation" for consideration by *eLife*. Your article has been reviewed by 3 peer reviewers, including Julie P I Welburn as the Reviewing Editor and Reviewer #1, and the evaluation has been overseen by a Senior Editor. The following individual involved in review of your submission has agreed to reveal their identity: Anne Straube (Reviewer #3).

Comments to the Authors:

We are sorry to say that, after consultation with the reviewers, we have decided that this work will not be considered further for publication in *eLife*.

Overall, the reviewers thought that some of the literature had been overlooked, where this phenotype has been previously reported and there were no mechanistic insights explaining the pathogenic role of the mutations of Kif22. However the reviewers thought the paper was very strong and with a little bit more biochemical experiments to dissect the function of the mutations on the motor and motor-tail regulation, this paper would be a good candidate for e*Life*.

*Reviewer #1:*

In this study, Thompson and colleagues characterize the pathogenic mutations in Kif22, some of which previously identified in patients and a new patient reported here. Using cell biology and quantitative image analysis, they show that polar ejection forces during mitosis are not compromised in the various mutants. Instead the mutants appear to cause defects in anaphase in nuclear envelope reformation and envelope solidty, separation of chromosomes and separation of the spindles. The reported phenotype is similar to that of CDK1-phosphomimetic Kif22. The phosphorylated state of kif22 has been previoulsy shown to be inactivated by dephosphorylation. Overall, this work supports a dominant negative mechanism of action of Kif22, where polar ejection forces need to be inactivated at the metaphase to anaphase transition to ensure completion of cell division and cytokinesis. The manuscript is well written and the figures are of high quality.

Based on human genetics, some cell types are more reliant than others on this inactivation. One drawback of this work presented here is that is uses HeLa cells rather than chondrocytes and speculate these cells would be the ones affected by the mutations. But the cell model already provides insights into the potential cause of Kif22 mutation related disorder. A outstanding question here is whether polar ejection forces mediated by Kid are linearly correlated with motor levels and activity. Would distinct levels of expression correlate with how far the chromosomes are ejected? Where maximal activity of the motor would correspond to maximal ejection? Another point is whether the region mutated normally regulates the motor activity? This would support their hypothesis.

The prediction method of pathogenicity is not well explained. Is there a pathogenic score associated the predicted phenotype for each predictor? Can the phenotype be better quantified and documented? It is hard to understand the x-ray pictures.

Some of the P-values are difficult to read, because there are so many bars on the graph. Figure S1D: can the scale be adjuted so we see a difference between the sample? They all look close to 0 so p-value not meaningful. Same with data presentation of Figure 1B.

Specify number of cells analyzed for each condition rather than give a range.

How is the motor activated and switched off? Does the region where mutations occur bind to the motor domains?

*Reviewer #2:*

This work identified a novel pathogenic mutation of kinesin KIF22 and characterized it together with the two previously known mutations associated with a developmental disorder spondyloepimetaphyseal dysplasia (SEMDJL2). The data revealed that these mutations disrupt inactivation of KIF22-mediated polar ejection forces (PEFs) in anaphase, leading to defects in chromosome segregation that disrupt cytokinesis in a subset of cells and consequently reduce the overall cell proliferation.

Overall, this is an interesting study that addresses an important question and offers an explanation for the role of KIF22 mutations in developmental disorders. The conclusions are well supported by the data, however some aspects have limited novelty and could be better discussed in light of previous literature. The study would benefit of a more mechanistic insight into how the described mutations prevent motor inactivation.

Strengths:

The study identified and characterized a novel pathogenic mutation in KIF22 and offered an explanation for the role of KIF22 mutations in developmental disorders. The work relies on a thorough experimental design accompanied with rigorous controls that included generation of multiple stable inducible cell lines, high quality quantification of cell phenotypes and detailed analysis of protein/mutants activity, localization and dynamics.

Weaknesses:

The work would need to be better discussed in light of previous literature. The first study showing that anaphase chromosomes recongress due to a failed inactivation of CDK1 and KIF22 has not been discussed in the context of presented data (Wolf et al., EMBO J 2006). Also, the effect of V475G mutant is not surprising, knowing that mutating 7 amino-acids within the same region (including V475) resulted in a similar anaphase recongression phenotype (Soeda et al., JCS 2016). The results of those two studies have indicated that PEFs, if not inactivated, are stronger than microtubule-sliding forces.

Some experiments were performed with a low cell number. For example, in Figure 5 only 2-3 cells were used per each of 3 independent experiments. This is likely the reason why, despite showing a clear trend, no statistical significance was observed under some conditions.

The study does not provide a more mechanistic insight into how the described mutations prevent motor inactivation.

A more mechanistic insight on how the mutations prevent KIF22 inactivation would greatly improve the manuscript. Could the head-tail autoinhibition and the microtubule-binding hypotheses discussed on page 46 be tested? Maybe something along these lines could be of help:

- Showing that the truncated versions of head and tail domains interact may provide a hint that the first hypothesis may be correct.

- Soeda et al. proposed that the second microtubule-binding domain in the tail of KIF22 contributes to inactivation of PEFs. Thus, one could predict that this domain prevents chromosomes to get stretched by PEFs. Is a similar stretching happening in KIF22 mutants presented in Figure 5H-I (the distance between centromeres and ends of chromosome arms increases)?

*Reviewer #3:*

The manuscript investigates mutations in the kinesin KIF22 that cause abnormal skeletal development. The main findings are that localisation, binding kinetics and generation of polar ejection forces during prometaphase are unaltered when comparing KIF22 mutants to the wildtype motor. However, cells expressing disease-causing mutants of KIF22 show defects in chromosome segregation due to a lack of inactivation of polar ejection forces during anaphase. This results in re-congression and in a significant subset of cells in cytokinesis failure and abnormal nuclear shape. All experiments have been conducted in HeLa cells with five different mutations, robustly quantified and reproduced in RPE1 cells with two key mutations. The authors then recapitulate the defects with a phosphomimetic T463D mutation, as phosphorylation of T463 has been shown previously to be required for generating polar ejection forces, and dephosphorylation at this residue was implicated in switching KIF22 off during anaphase. All mutations acted dominant in the experiment and also in patients, consistent with finding a gain-of-function defect. The finding that mutations in the motor domain and the tail cause similar defects suggests that an interaction of motor domain and tail might be involved in switching off KIF22's polar ejection force generation at the onset of anaphase and thus offers a testable hypothesis for future studies.

As the idea that polar ejection force needs to be switched off during anaphase is not new, the impact of the study for understanding cell division is somewhat limited. The authors discuss possible mechanisms why mutations in the motor domain and the tail show similar effects, but no experiments have been included to test these ideas. For example, testing for the proposed interaction of motor domain and tail domain would be a relatively straightforward biochemistry experiment and offer additional mechanistic insights that would increase the impact of the paper and also enable providing an explanation why these specific mutations cause disease if such motor-tail interactions would be reduced in the presence of the mutations.

While the paper shows that defects occur only in anaphase, localisation and turnover have only been analysed in metaphase. It would make sense to compare localisation and turnover in anaphase. This might also offer some insights into the switching off mechanism.

There isn't currently an explanation why such defects in cell division cause tissue-specific defects in skeletal development as cell division was investigated here only with cell types not involved in bone growth. As the manuscripts describes a new patient, I wonder whether the authors have access to biopsies that might allow showing nuclear deformations in patient tissue and whether this occurs in a tissue-specific manner.

Most cell biologists won't be familiar with normal bone images. Thus it would be valuable to add control images from a healthy person of similar age to illustrate the patient symptoms in (Figure 1F,G).

[Editors’ note: further revisions were suggested prior to acceptance, as described below.]

Thank you for resubmitting your work entitled "Pathogenic mutations in the chromokinesin KIF22 disrupt anaphase chromosome segregation" for further consideration by *eLife*. Your revised article has been evaluated by Anna Akhmanova (Senior Editor) and a Reviewing Editor.

The manuscript has been improved but there are some remaining issues that need to be addressed, as outlined below:

1) The negative data showing the tail and motor do not interact should be added to the paper and discussed in the paper.

2) There is no direct evidence that this site is differentially phosphorylated during mitosis (Prometaphase and metaphase vs. Anaphase), and the Alanine mutant is fully functional, indicating that loss of this putative phosphorylation site is not sufficient to induce inactivation. Therefore it is equally possible that introduction of a charge disrupts the structure of the coiled-coil domain, thus mimicking the deletion of this domain. Please clarify the manuscript to highlight this region is important for Kif22 activity but not necessarily phosphorylated.

3) Some experimental controls are missing and should be added, showing the expression levels and knockdown efficiency in the cell lines used.

4) Please edit the manuscript about the interpretation of cell proliferation data to avoid overinterpretation. The authors assume in their model two extreme possibilities: either that nuclear architecture defects or that cytokinesis defects are entirely responsible for the reduced proliferation. It could, however, be that both types of defects lead to a slow down of the cell cycle, which would also explain the reduced proliferation rate. If the authors want to claim that cytokinesis failure is the primary reason for a reduced proliferation, they would need to monitor cell populations over long time periods and demonstrate that the cell doubling time of cells with failed cytokinesis is substantially longer than the rest of the population (this is likely, but at this stage not shown).

*Reviewer #1 (Recommendations for the authors):*

The authors have addressed most comments from the last round of revisions well.

One of the major points raised by the reviewers was to test whether the tail inhibited the motor. The authors have tested this and they do not see any interaction between the tail and motor that would inhibit/activate the kinesin using coIP and various approaches. So no clear mechanism for how Kif22 might be switched on and off emerge. Of course the paper then does not explain how Kif22 is working. However with the current data and all the studied mutants, the authors debate different models for the regulation of Kif22.

I think understanding the mechanism of Kif22 regulation would require a lot of biochemistry work, which is a different paper and not in the expertise of the authors. They tried to address all the comments and some experiments supporting tail-motor regulation we asked did not work technically, but some phosphomutants are further characterised. The negative data on the tail-motor showing they do not interact should be integrated into the manuscript.

*Reviewer #2 (Recommendations for the authors):*

I would like to congratulate the authors for a great work done during the revision of this manuscript. I appreciate their efforts to address the mechanism underlying the motor inactivation and find the discussion part covering that topic largely improved. The added experiments exploring the phosphoregulation of the α-2 helix show that the mutation-bearing region regulates motor activity. This is an important finding that significantly improves the manuscript. All my concerns were properly addressed and I recommend the revised manuscript for publication.

*Reviewer #3 (Recommendations for the authors):*

This revised manuscript investigates the functional consequences of KIF22 mutations appearing in human pathologies during cell division. The authors conclude that these mutations do not affect the ability of KIF22 to generate a polar ejection force, but rather prevent the inactivation of KIf22 at anaphase onset, resulting in chromosome recongression, and in the extreme case to cytokinesis failure due to chromosomes masses in the site of cell division.

The study investigates an interesting link between a human pathology and a cell biology mechanism. It is therefore novel and original and of interest to a wider public. The cell biology scope of the study remains limited, since the fact that KIF22 must be inactivated at anaphase onset to prevent chromosome segregation defects, was already known. The weakness of the paper is that beyond this tight correlation between pathological mutations and anaphase defects, the authors do not offer major mechanistic advances in the inactivation mechanisms of KIF22, a point that remains unaddressed even after the revision work.

Indeed, even though the authors have done a good job at addressing many of the points of the reviewers, they could not obtain a major mechanistic understanding in how the α-2-helix of Kif22 or could be involved in the inactivation of KIF22. The additional characterization of the T463 mutation offers more support for a role of the coiled-coil domain, but I am not convinced that the presented data prove that this threonine must dephosphorylated to allow inactivation of KIF22: there is no direct evidence that this site is differentially phosphorylated during mitosis (Prometaphase and metaphase vs. Anaphase), and the Alanine mutant is fully functional, indicating that loss of this putative phosphorylation site is not sufficient to induce inactivation. Therefore it is equally possible that introduction of a charge disrupts the structure of the coiled-coil domain, thus mimicking the deletion of this domain.

Finally, I would suggest that the authors are more cautious when interpreting the cell proliferation data. The authors assume in their model two extreme possibilities: either that nuclear architecture defects or that cytokinesis defects are entirely responsible for the reduced proliferation. It could, however, be that both types of defects lead to a slow down of the cell cycle, which would also explain the reduced proliferation rate. If the authors want to claim that cytokinesis failure is the primary reason for a reduced proliferation, they would need to monitor cell populations over long time periods and demonstrate that the cell doubling time of cells with failed cytokinesis is substantially longer than the rest of the population (this is likely, but at this stage not shown).

---

## [Author Response]

[Editors’ note: the authors resubmitted a revised version of the paper for consideration. What follows is the authors’ response to the first round of review.]

Reviewer #1:In this study, Thompson and colleagues characterize the pathogenic mutations in Kif22, some of which previously identified in patients and a new patient reported here. Using cell biology and quantitative image analysis, they show that polar ejection forces during mitosis are not compromised in the various mutants. Instead the mutants appear to cause defects in anaphase in nuclear envelope reformation and envelope solidty, separation of chromosomes and separation of the spindles. The reported phenotype is similar to that of CDK1-phosphomimetic Kif22. The phosphorylated state of kif22 has been previoulsy shown to be inactivated by dephosphorylation. Overall, this work supports a dominant negative mechanism of action of Kif22, where polar ejection forces need to be inactivated at the metaphase to anaphase transition to ensure completion of cell division and cytokinesis. The manuscript is well written and the figures are of high quality.Based on human genetics, some cell types are more reliant than others on this inactivation. One drawback of this work presented here is that is uses HeLa cells rather than chondrocytes and speculate these cells would be the ones affected by the mutations. But the cell model already provides insights into the potential cause of Kif22 mutation related disorder.A outstanding question here is whether polar ejection forces mediated by Kid are linearly correlated with motor levels and activity. Would distinct levels of expression correlate with how far the chromosomes are ejected? Where maximal activity of the motor would correspond to maximal ejection?

We appreciate this question, and measured GFP expression level in cells from which polar ejection force measurements were obtained to answer it. GFP expression level does not correlate with the distance of the chromosomes from the spindle pole in monopolar spindles, suggesting increased expression of wild type or mutant KIF22 does not necessarily lead to increased polar ejection force in prometaphase. These data are presented in Figure 3E (in the presence of endogenous KIF22) and 3F (KIF22 knockdown).

Another point is whether the region mutated normally regulates the motor activity? This would support their hypothesis.

This a great question, and one we endeavored to answer by considering whether posttranslational modification of the α-2 helix may regulate KIF22 inactivation, analogous to the reported phosphoregulation of KIF22 inactivation at T463 in the tail (Soeda et al. 2016). We tested the consequences of mimicking the phosphorylation of two α-2 helix residues previously identified in phosphoproteomic screens, T134 and T158.

Both phosphomimetic and phosphonull mutations at T134 disrupt the localization of KIF22. It accumulates on spindle microtubules, consistent with previous use of T134N as a rigor mutation (Tokai-Nishizumi 2005). This is not consistent with this amino acid regulating KIF22 inactivation. These results are documented in a new figure, Supplemental Figure 6.

We determined that mimicking phosphorylation of T158 in the α-2 helix does, in fact, phenocopy the effects of both pathogenic mutations and the tail phosphomimetic T463D mutation. Cells expressing KIF22-GFP T158D demonstrate anaphase chromosome recongression and nuclear morphology defects. Importantly, expression of KIF22-T158A does not induce these phenotypes. These results are documented in a new figure, Figure 9. The T158 data suggest that yes, the α-2 helix region where pathogenic mutations were identified does regulate motor activity not only in SEMDJL2 but also outside of the context of human disease.

The prediction method of pathogenicity is not well explained. Is there a pathogenic score associated the predicted phenotype for each predictor?

We have added scores for predictions of variant significance to Supplemental Table 1.

Can the phenotype be better quantified and documented? It is hard to understand the x-ray pictures.

We appreciate the challenge of interpreting the patient radiographs and have revised the text (lines 142-145), added an arrowhead to Figure 1F to highlight the fourth metacarpal, and revised the figure legends for Figure 1F and 1G. A quantitative comparison of skeletal phenotypes between patients isn’t possible as only one patient has been identified with the V475G mutation.

Some of the P-values are difficult to read, because there are so many bars on the graph.

We have increased the font size of the p-values in all figures.

Figure S1D: can the scale be adjuted so we see a difference between the sample? They all look close to 0 so p-value not meaningful. Same with data presentation of Figure 1B.

We have added additional graphs presenting the data in Figure S1B and S1D with adjusted y-axes to clearly show variability between experimental groups (right graphs). We have also kept the data as originally presented in these panels (left graphs), as the axes match those in figures S1C and S1E to facilitate comparison of expression levels between treatments.

Specify number of cells analyzed for each condition rather than give a range.

We have edited all figure legends to list the number of cells and number of experimental replicates for each condition rather than using ranges.

How is the motor activated and switched off? Does the region where mutations occur bind to the motor domains?

We agree that the question of how KIF22 is inactivated is a very interesting one. In response to reviewer comments, we tested whether we could observe an interaction between the motor domain and tail of KIF22.

To do this, we built a cell line with inducible expression of a GFP-tagged KIF22 motor domain construct (amino acids 1-383) and built plasmids for transient transfection to express mCherry-tagged KIF22 tail fragments. One mCh-Tail plasmid encoded a shorter construct containing only the described microtubule binding and coiled-coil domains (amino acids 442-506), and one plasmid encoded a slightly longer construct also containing the microtubule binding and coiled-coil tail domains (amino acids 420-520). Both constructs did not include the two nuclear localization signals in the KIF22 tail, and mCh-Tail localized to the cytoplasm in interphase and mitotic cells. Interestingly, no localization to the microtubules was observed in interphase or mitotic cells expressing either mCh-Tail construct, despite the presence of the microtubule binding domain in both.

We performed co-immunoprecipitations using lysates from cells expressing Motor Domain-GFP and mCh-Tail to test for an interaction between these domains of KIF22. While we were able to IP Motor Domain-GFP or mCh-Tail, we did not observe co-IP of the motor domain with the tail. We performed reciprocal experiments immunoprecipitating either Motor Domain-GFP or mCh-Tail and did not observe co-IP in either case. Additionally, given the role of phosphorylation of tail residue T463 in regulating KIF22 inactivation, we treated cells with the CDK inhibitor RO-3306 to reduce phosphorylation of T463. However, we did not observe co-IP of mCh-Tail with Motor Domain-GFP in lysates from cells treated with the CDK inhibitor.

These data suggest that a strong, long-lasting physical interaction between the tail and motor domain of KIF22 did not likely occur under the conditions of our assays. However, they do not rule out that such an interaction may be weak or dynamic and, therefore, not detectable using this assay. Similarly, our observation that KIF22 tail fragments do not localize to microtubules in interphase, metaphase, or anaphase are not consistent with the model of microtubule binding-mediated KIF22 inactivation (Soeda et al. 2016). However, the data do not preclude weak or dynamic interactions between the tail and the microtubule surface contributing to the inactivation of KIF22. It is also possible that the presence of the motor domain is necessary for strong interactions of the KIF22 tail with the microtubule surface and consequent motor inactivation.

Given that our negative IP results cannot rule out an interaction between the tail and motor domain, and the large parameter space that could still be explored to detect a potential interaction (e.g. changes to motor and tail constructs, different buffers, locations of tags) we prefer not to include the IP data in the manuscript. We believe that the reviewers’ suggestion to test the head-tail interaction model was fair. However, our data indicate that this will not be a simple problem to solve. We feel that a study of potential conformational changes to KIF22 at the metaphase-to-anaphase transition is likely worthy of a complete paper and is thus beyond the scope of the current work.

In the revised discussion, we present our data in the context of the previously proposed model for KIF22 inactivation (Soeda et al. 2016) and additionally offer that physical tailmotor domain interactions or allosteric effects of the tail on the motor domain may contribute to the inactivation of KIF22. These models are not mutually exclusive, and our data suggest that regulation of KIF22 activity may not occur via a single, simple mechanism.

Our work is the first to demonstrate the involvement of the motor domain of KIF22 in inactivation. This conclusion is supported by data indicating that pathogenic mutations in the α-2 helix disrupt inactivation and that mimicking phosphorylation of T158 in the α-2 helix also disrupts inactivation. Further studies of this mechanism will need to consider how the tail, the motor domain, and the microtubules interact to inactivate KIF22.

Reviewer #2:This work identified a novel pathogenic mutation of kinesin KIF22 and characterized it together with the two previously known mutations associated with a developmental disorder spondyloepimetaphyseal dysplasia (SEMDJL2). The data revealed that these mutations disrupt inactivation of KIF22-mediated polar ejection forces (PEFs) in anaphase, leading to defects in chromosome segregation that disrupt cytokinesis in a subset of cells and consequently reduce the overall cell proliferation.Overall, this is an interesting study that addresses an important question and offers an explanation for the role of KIF22 mutations in developmental disorders. The conclusions are well supported by the data, however some aspects have limited novelty and could be better discussed in light of previous literature. The study would benefit of a more mechanistic insight into how the described mutations prevent motor inactivation.Strengths:The study identified and characterized a novel pathogenic mutation in KIF22 and offered an explanation for the role of KIF22 mutations in developmental disorders. The work relies on a thorough experimental design accompanied with rigorous controls that included generation of multiple stable inducible cell lines, high quality quantification of cell phenotypes and detailed analysis of protein/mutants activity, localization and dynamics.Weaknesses:The work would need to be better discussed in light of previous literature. The first study showing that anaphase chromosomes recongress due to a failed inactivation of CDK1 and KIF22 has not been discussed in the context of presented data (Wolf et al., EMBO J 2006).

We thank the reviewer for highlighting this study. We agree that it is important to reference. We now discuss it in the introduction (lines 65-67), results (lines 378-380), and discussion (lines 914-917) sections of the paper.

Also, the effect of V475G mutant is not surprising, knowing that mutating 7 amino-acids within the same region (including V475) resulted in a similar anaphase recongression phenotype (Soeda et al., JCS 2016).

We thank the reviewer for encouraging us to better discuss our findings in the context of the work of Soeda et al. We agree that this paper is important for interpreting our results and cite it throughout the introduction and discussion. The assays in the Soeda 2016 study included assessment of tail domain deletions and mutation of 7 or 8 coiled-coil domain amino acids to serine or aspartate, respectively. While our V475G findings are consistent with their observations of cells with 7S or 8D mutations in the coiled-coil domain, we felt it was important to characterize the effect of a single point mutation rather than a larger alteration of tail structure, as the effects of a single mutation may not have matched the effects of these larger-scale changes.

The results of those two studies have indicated that PEFs, if not inactivated, are stronger than microtubule-sliding forces.

We appreciate the opportunity to clarify how our results differ from previous published studies. In Wolf et al. 2006, chromosome movements in cells with nondegradable cyclin B1 are assessed using H2B-GFP. This work does not use markers for the spindle poles or centromeres to assess the movements of these structures. The work of Wolf et al. includes imaging of microtubules and kinetochores in fixed cells, and the conclusions drawn from these experiments are that in cells with nondegradable cyclin B1, spindles are bipolar and kinetochores are not paired during recongression. For cells subjected to microinjection of anti-KIF22 neutralizing antibodies, the authors focus on chromosome movements, again using H2B-GFP as a marker for live imaging.

The results of the experiments presented by Wolf et al. are consistent with, but distinct from, our work. We used pericentrin-RFP and CENPB-mCh to image the movements of the spindle poles and kinetochores in live cells with high temporal resolution to directly compare the movements of these structures to chromosome movements during anaphase recongression. This approach allowed us to assess the effects of a failure to inactivate KIF22 on anaphase A and anaphase B. Our work is the first to demonstrate that anaphase polar ejection forces affect anaphase A by altering the movements of chromosome arms, but not the shortening of k-fibers (Figure 5H and 5I) and affect anaphase B by altering spindle pole separation (Figure 5F and 5G).

The movement of the spindle during anaphase recongression caused by deletion of the tail microtubule binding or coiled-coil domains was assessed by Soeda et al. using mChtubulin. They did not observe differences in spindle length during anaphase in cells with anaphase polar ejection forces, and in fact conclude that these forces do not affect spindle elongation. This is different than our result that the distance between spindle poles is reduced by anaphase polar ejection forces (Figure 5F and 5G), and we discuss this in the manuscript (lines 1019-1021). This difference may be due to the fact that Soeda et al. were observing meiosis in mouse oocytes.

Thus, neither the work presented in Wolf et al. 2006 nor Soeda et al. 2016 demonstrates that anaphase polar ejection forces overcome microtubule sliding forces to affect the movements of spindle poles (anaphase B).

Some experiments were performed with a low cell number. For example, in Figure 5 only 2-3 cells were used per each of 3 independent experiments. This is likely the reason why, despite showing a clear trend, no statistical significance was observed under some conditions.

We appreciate the reviewer’s interest in ensuring sample sizes are large enough to draw robust conclusions. The assays in Figure 5 require the identification of metaphase cells expressing both pericentrin-RFP and CENPB-mCh at appropriate levels and that the cells divide parallel to the imaging plane so that both spindle poles remain in focus. These challenges mean that throughput for this assay is quite low – the data originally presented was from six independent experiments, but some experiments included zero cells expressing one construct type that were ultimately suitable for analysis, hence the presented data included cells from three experiments per condition.

We repeated this experiment three additional times and were able to analyze 3 cells expressing KIF22-GFP, 1 cell expressing KIF22-GFP R149Q, and 3 cells expressing KIF22-GFP V475G from these additional experimental replicates. Including these new data, the reduction in distance between chromosome masses (Figure 5C), centromeres (Figure 5E), and spindle poles (Figure 5G) 10 minutes after anaphase onset is now significant for cells expressing KIF22-GFP R149Q or KIF22-GFP V475G.

The study does not provide a more mechanistic insight into how the described mutations prevent motor inactivation.

We agree that the question of how KIF22 is inactivated is an interesting one.

To provide additional mechanistic insight into KIF22 inactivation, we explored whether the α-2 helix of the motor domain is required for this process outside of the context of disease-derived mutations. We demonstrated that mimicking phosphorylation of T158 in α-2 causes anaphase chromosome recongression and consequent nuclear morphology defects, suggesting that the α-2 helix contributes to inactivation of KIF22. Our work is the first to demonstrate a role for the motor domain in KIF22 inactivation. Previous work (Soeda et al. 2016) focused on the necessity of the microtubule binding and coiled-coil domains found in the tail, as well as T463 dephosphorylation, for KIF22 inactivation.

Identifying the motor domain as a component of KIF22 inactivation informs our understanding of the physical mechanism of KIF22 inactivation. We discuss our data in the context of the previously proposed model for KIF22 inactivation (Soeda et al. 2016) and additionally offer that physical tail-motor domain interactions or allosteric effects of the tail on the motor domain may contribute to the inactivation of KIF22. These models are not mutually exclusive, and our data suggest that regulation of KIF22 activity may not occur via a single, simple mechanism.

A more mechanistic insight on how the mutations prevent KIF22 inactivation would greatly improve the manuscript. Could the head-tail autoinhibition and the microtubule-binding hypotheses discussed on page 46 be tested? Maybe something along these lines could be of help:- Showing that the truncated versions of head and tail domains interact may provide a hint that the first hypothesis may be correct.

Please see our comments on testing mechanisms of KIF22 autoinhibition in response to Reviewer #1’s comment above. Briefly, we expressed Motor Domain-GFP and mCherryTail and used co-immunoprecipitation to test for an interaction between these domains of KIF22. We were unable to detect an interaction between the tail and motor domain of KIF22 under these conditions. However, these negative results do not rule out that such an interaction may be weak or dynamic and, therefore, not detectable using this assay. We also observed that KIF22 tail fragments do not localize to microtubules in interphase, metaphase, or anaphase. These data are inconsistent with the model of microtubule binding-mediated KIF22 inactivation (Soeda 2016), but similar to our co-IP results, also do not preclude weak or dynamic interactions between the tail and the microtubule surface contributing to the inactivation of KIF22. It is also possible that the presence of the motor domain is necessary for strong interactions of the KIF22 tail with the microtubule surface and consequent motor inactivation.

- Soeda et al. proposed that the second microtubule-binding domain in the tail of KIF22 contributes to inactivation of PEFs. Thus, one could predict that this domain prevents chromosomes to get stretched by PEFs. Is a similar stretching happening in KIF22 mutants presented in Figure 5H-I (the distance between centromeres and ends of chromosome arms increases)?

We appreciate the reviewer’s interest in how our data on chromosome arm and centromere positions relative to the spindle poles in anaphase may inform our understanding of the mechanism of KIF22 inactivation. Our data do show that the chromosome arms are pushed further from the poles in anaphase in cells expressing mutant KIF22 (Figure 5H), while centromere distance from the poles is not affected (Figure 5I). We interpret this as consistent with anaphase recongression being caused by increased polar ejection forces on chromosome arms. We are not able to specifically measure the distance between the centromere and the ends of chromosome arms for individual chromosomes given the density of chromosome masses in HeLa-Kyoto cells.

Reviewer #3:The manuscript investigates mutations in the kinesin KIF22 that cause abnormal skeletal development. The main findings are that localisation, binding kinetics and generation of polar ejection forces during prometaphase are unaltered when comparing KIF22 mutants to the wildtype motor. However, cells expressing disease-causing mutants of KIF22 show defects in chromosome segregation due to a lack of inactivation of polar ejection forces during anaphase. This results in re-congression and in a significant subset of cells in cytokinesis failure and abnormal nuclear shape. All experiments have been conducted in HeLa cells with five different mutations, robustly quantified and reproduced in RPE1 cells with two key mutations. The authors then recapitulate the defects with a phosphomimetic T463D mutation, as phosphorylation of T463 has been shown previously to be required for generating polar ejection forces, and dephosphorylation at this residue was implicated in switching KIF22 off during anaphase. All mutations acted dominant in the experiment and also in patients, consistent with finding a gain-of-function defect. The finding that mutations in the motor domain and the tail cause similar defects suggests that an interaction of motor domain and tail might be involved in switching off KIF22's polar ejection force generation at the onset of anaphase and thus offers a testable hypothesis for future studies.As the idea that polar ejection force needs to be switched off during anaphase is not new, the impact of the study for understanding cell division is somewhat limited.

We agree with the reviewer that previous work (Wolf et al. 2006, Su et al. 2016, Soeda et al. 2016) demonstrates that a failure to reduce polar ejection forces in anaphase can disrupt chromosome segregation. We have discussed these data throughout the revised manuscript, including in the introduction (lines 65-82). We have also attempted to clarify gaps in our understanding of how polar ejections forces are reduced during anaphase and why this is necessary for proper chromosome segregation to occur. In summary, the novel findings we present include:

Characterization of the consequences of a failure to inactivate polar ejection forces in cells: we show not only that anaphase polar ejection forces can disrupt chromosome movements, but also that daughter cell phenotypes are altered in cells experiencing aberrant polar ejection forces. We demonstrate that daughter cells have abnormally shaped nuclei (Figures 6 and S4), that the nuclear envelopes of these nuclei are capable of retaining mCh-NLS (Figure 6G), that cytokinesis failure occurs in about 30% of cells (Figure 4D and 4E), that cell proliferation is reduced, and that this reduction in proliferation is consistent with the rate of cytokinesis failure limiting the rate of proliferation (Figure 7). The only previous study to address this question is Wolf et al. 2006, which includes an observation that a number of cells expressing nondegradable cyclin B1 failed to form two daughter cells.

The effects of polar ejection forces on both anaphase A and anaphase B: as described in our response to reviewer #2 above, our work is the first to demonstrate that a failure to inactivate KIF22 in anaphase affects the movement of the spindle poles, suggesting that the polar ejection forces collectively generated by mutant KIF22 motors are of greater magnitude than the forces sliding the spindle poles apart during anaphase B.

The involvement of the motor domain α-2 helix in KIF22 inactivation: we show that pathogenic mutations at P148 or R149 or mimicking phosphorylation of T158 disrupts anaphase chromosome segregation. This implicates the α-2 helix in KIF22 inactivation for the first time, adding to previous work (Soeda et al. 2016) that defines the requirement of tail domains for polar ejection force suspension.

Connecting anaphase polar ejection forces to human health and development: while further study is needed to connect the cellular phenotypes we present to patient pathology, our characterization of the cellular consequences of pathogenic mutations in KIF22 provides the first potential connection between a failure to suspend polar ejection forces in anaphase and human pathology.

The authors discuss possible mechanisms why mutations in the motor domain and the tail show similar effects, but no experiments have been included to test these ideas. For example, testing for the proposed interaction of motor domain and tail domain would be a relatively straightforward biochemistry experiment and offer additional mechanistic insights that would increase the impact of the paper and also enable providing an explanation why these specific mutations cause disease if such motor-tail interactions would be reduced in the presence of the mutations.

We appreciate the interest of all three reviewers in the mechanism of KIF22 inactivation. Please see our comments on testing mechanisms of KIF22 autoinhibition in response to Reviewer #1’s comment above. Briefly, we expressed Motor Domain-GFP and mCherry-Tail and used co-immunoprecipitations to test for an interaction between these domains of KIF22. We were unable to detect a physical interaction between the tail and motor domain of KIF22. However, these negative results do not rule out that such an interaction may be weak or dynamic and, therefore, not detectable using this assay. Similarly, we observed that KIF22 tail fragments do not localize to microtubules in interphase, metaphase, or anaphase. These data are inconsistent with a strong tailmicrotubule interaction that would support the model of microtubule binding-mediated KIF22 inactivation (Soeda et al. 2016), but also do not preclude weak or dynamic interactions between the tail and the microtubule surface contributing to the inactivation of KIF22. It is also possible that the presence of the motor domain is necessary for strong interactions of the KIF22 tail with the microtubule surface and consequent motor inactivation.

We agree that additional biochemical and biophysical studies of KIF22 would be of interest. However, to our knowledge, soluble, active KIF22 has not been purified in quantities required for these types of studies. There are very few studies of purified KIF22 for this reason, and those that exist have utilized low yield purifications for single molecule studies. Additionally, the lack of in vitro data on KIF22 means that there are fundamental questions that remain unanswered about this motor, including its oligomeric state. It has been characterized as a monomer by analytical ultracentrifugation (Yajima et al. 2003, Shiroguchi et al. 2003), but plus-end-directed movement of GFP-KIF22 oligomers on microtubules has been observed using single molecule TIRF (Stumpff et al. 2012). These technical and interpretation challenges limit our ability to perform biochemical or biophysical experiments to further characterize mutations in KIF22.

While the paper shows that defects occur only in anaphase, localisation and turnover have only been analysed in metaphase. It would make sense to compare localisation and turnover in anaphase. This might also offer some insights into the switching off mechanism.

We agree with the reviewer that altered localization or turnover would be an interesting potential mechanism of KIF22 inactivation in anaphase. This question is what prompted us to assess localization and turnover in anaphase. Our results demonstrate that localization and dynamics of KIF22 are not altered by pathogenic mutations during interphase, metaphase, or anaphase. The localization of KIF22-GFP can be seen in the live imaging presented in Figures 4A and S3D, and FRAP data demonstrating that exchange of KIF22-GFP, KIF22-GFP R149Q, and KIF22-GFP V475G is low in anaphase is presented in Figures 2D, 2G, and 2J, respectively. These data, which were included in both the original and revised manuscripts, demonstrate that inactivation of KIF22 in anaphase does not alter the localization or turnover of the motor.

There isn't currently an explanation why such defects in cell division cause tissue-specific defects in skeletal development as cell division was investigated here only with cell types not involved in bone growth. As the manuscripts describes a new patient, I wonder whether the authors have access to biopsies that might allow showing nuclear deformations in patient tissue and whether this occurs in a tissue-specific manner.

We thank the reviewer for highlighting this important question. We agree that patient samples could help connect cellular phenotypes to organismal pathology. However, we do not have access to patient samples for these tests. Access to patient samples is limited by the very small number of people diagnosed with SEMDJL2 and the fact that most patients are adolescents. We discuss the question of tissue specificity in the manuscript (lines 1022-1044) and plan to address this question more directly in future studies.

Most cell biologists won't be familiar with normal bone images. Thus it would be valuable to add control images from a healthy person of similar age to illustrate the patient symptoms in (Figure 1F,G).

We appreciate the challenge of interpreting the patient radiographs and have revised the text (lines 142-145), added an arrowhead to Figure 1F to highlight the fourth metacarpal, and revised the figure legends for Figure 1F and 1G. Unfortunately, we do not have the ability to identify an individual to serve as a paired control for additional imaging.

[Editors’ note: what follows is the authors’ response to the second round of review.]

The manuscript has been improved but there are some remaining issues that need to be addressed, as outlined below:1) The negative data showing the tail and motor do not interact should be added to the paper and discussed in the paper.

We appreciate the interest of the reviewers in these data and have added them in a new supplemental figure (Figure 9 —figure supplement 3). We have included images of the localization of mCh-Tail fragments and immunoprecipitation data testing the interaction of Motor Domain-GFP and mCh-Tail fragments. These data are described in the results (lines 848-862) and discussion (lines 1035-1038) sections of the manuscript. We have included a statement in the discussion that further studies would be needed to rule out a physical inactivating interaction between these domains of KIF22.

2) There is no direct evidence that this site is differentially phosphorylated during mitosis (Prometaphase and metaphase vs. Anaphase), and the Alanine mutant is fully functional, indicating that loss of this putative phosphorylation site is not sufficient to induce inactivation. Therefore it is equally possible that introduction of a charge disrupts the structure of the coiled-coil domain, thus mimicking the deletion of this domain. Please clarify the manuscript to highlight this region is important for Kif22 activity but not necessarily phosphorylated.

We agree that studies determining the phosphorylation status of T158 in different phases of mitosis are warranted to understand if or when phosphorylation of this residue regulates KIF22 activity and have added language to the Discussion section that states this (lines 1003-1007). We have emphasized in the discussion that our work examining charge changes at T158 support a role for the motor domain in KIF22 inactivation.

T158 is located in the a2 helix of the motor domain. To consider whether charge change at T158 may disrupt the structure of this helix, we used ColabFold (Mirdita et al. 2022, PMID 35637307) to predict the structure of the KIF22 T158D motor domain. No disruption of the a2 helix was predicted. We do not think it is likely that the T158D phosphomimetic mutation affects KIF22 function by disrupting the structure of this helix. Given the conservation and important mechanochemical role of this helix across kinesins, we would also predict that a complete disruption of this region would lead to loss of motor activity rather than constitutive activation.

The previously characterized phosphorylation site, T463, is located in the tail of KIF22, near the coiled-coil domain. This site is phosphorylated in mitosis, and its dephosphorylation occurs with similar kinetics to degradation of cyclin B, indicating that it is dephosphorylated as cells transition from metaphase to anaphase (Ohsugi et al. 2003, PMID 12727876). Additional studies demonstrate that mimicking phosphorylation of T463 or deletion or disruption of the coiled-coil domain cause anaphase chromosome recongression (Soeda et al. 2016, PMID 27550518). As the reviewer points out, whether or how phosphorylation of T463 affects the coiled-coil domain is unknown. We have added a note to the Discussion section stating that charge changes at T463 may disrupt the structure of tail domains necessary for KIF22 inactivation (lines 1013-1014).

3) Some experimental controls are missing and should be added, showing the expression levels and knockdown efficiency in the cell lines used.

We have added data measuring KIF22 expression levels and knockdown efficiency in T134 cell lines, included in a revised supplemental figure (Figure 9 —figure supplement 1). We have added similar data for T158 cell lines in a new supplemental figure (Figure 9 —figure supplement 2). We also demonstrate that anaphase recongression depends on expression level in the cell lines used to assess T158 (Figure 9 —figure supplement 2 E), and that the effect of T158D on recongression is also observed when only cells with low KIF22-GFP expression levels are considered (Figure 9 —figure supplement 2 F and G).

4) Please edit the manuscript about the interpretation of cell proliferation data to avoid overinterpretation. The authors assume in their model two extreme possibilities: either that nuclear architecture defects or that cytokinesis defects are entirely responsible for the reduced proliferation. It could, however, be that both types of defects lead to a slow down of the cell cycle, which would also explain the reduced proliferation rate. If the authors want to claim that cytokinesis failure is the primary reason for a reduced proliferation, they would need to monitor cell populations over long time periods and demonstrate that the cell doubling time of cells with failed cytokinesis is substantially longer than the rest of the population (this is likely, but at this stage not shown).

We appreciate the careful assessment of our data by the reviewers and have added language to the Results section clarifying that both nuclear morphology defects and cytokinesis failure may contribute to reduced proliferation (lines 619-621).

Reviewer #1 (Recommendations for the authors):The authors have addressed most comments from the last round of revisions well.One of the major points raised by the reviewers was to test whether the tail inhibited the motor. The authors have tested this and they do not see any interaction between the tail and motor that would inhibit/activate the kinesin using coIP and various approaches. So no clear mechanism for how Kif22 might be switched on and off emerge. Of course the paper then does not explain how Kif22 is working. However with the current data and all the studied mutants, the authors debate different models for the regulation of Kif22.I think understanding the mechanism of Kif22 regulation would require a lot of biochemistry work, which is a different paper and not in the expertise of the authors. They tried to address all the comments and some experiments supporting tail-motor regulation we asked did not work technically, but some phosphomutants are further characterised. The negative data on the tail-motor showing they do not interact should be integrated into the manuscript.

Please see response to point 1, above.

Reviewer #3 (Recommendations for the authors):This revised manuscript investigates the functional consequences of KIF22 mutations appearing in human pathologies during cell division. The authors conclude that these mutations do not affect the ability of KIF22 to generate a polar ejection force, but rather prevent the inactivation of KIf22 at anaphase onset, resulting in chromosome recongression, and in the extreme case to cytokinesis failure due to chromosomes masses in the site of cell division.The study investigates an interesting link between a human pathology and a cell biology mechanism. It is therefore novel and original and of interest to a wider public. The cell biology scope of the study remains limited, since the fact that KIF22 must be inactivated at anaphase onset to prevent chromosome segregation defects, was already known. The weakness of the paper is that beyond this tight correlation between pathological mutations and anaphase defects, the authors do not offer major mechanistic advances in the inactivation mechanisms of KIF22, a point that remains unaddressed even after the revision work.Indeed, even though the authors have done a good job at addressing many of the points of the reviewers, they could not obtain a major mechanistic understanding in how the α-2-helix of Kif22 or could be involved in the inactivation of KIF22. The additional characterization of the T463 mutation offers more support for a role of the coiled-coil domain, but I am not convinced that the presented data prove that this threonine must dephosphorylated to allow inactivation of KIF22: there is no direct evidence that this site is differentially phosphorylated during mitosis (Prometaphase and metaphase vs. Anaphase), and the Alanine mutant is fully functional, indicating that loss of this putative phosphorylation site is not sufficient to induce inactivation. Therefore it is equally possible that introduction of a charge disrupts the structure of the coiled-coil domain, thus mimicking the deletion of this domain.

Please see response to point 2, above. We agree that the time course of motor domain residue T158 phosphorylation during mitosis is unknown and have emphasized this point in our revised manuscript (lines 1003-1007). Published results supporting the role of phosphorylation of tail residue T463 in the regulation of KIF22 inactivation include observations that T463 is phosphorylated by CDK1 (Ohsugi et al. 2003, PMID 12727876), that allowing CDK1 activity in anaphase via expression of non-degradable cyclin B1 disrupts chromosome segregation and that this disruption depends on KIF22 activity (Wolf et al. 2006, PMID 16724106), and that mimicking phosphorylation of T463 also disrupts anaphase chromosome segregation (Soeda et al. 2016, PMID 27550518).

Finally, I would suggest that the authors are more cautious when interpreting the cell proliferation data. The authors assume in their model two extreme possibilities: either that nuclear architecture defects or that cytokinesis defects are entirely responsible for the reduced proliferation. It could, however, be that both types of defects lead to a slow down of the cell cycle, which would also explain the reduced proliferation rate. If the authors want to claim that cytokinesis failure is the primary reason for a reduced proliferation, they would need to monitor cell populations over long time periods and demonstrate that the cell doubling time of cells with failed cytokinesis is substantially longer than the rest of the population (this is likely, but at this stage not shown).

Please see response to point 4, above.